# Natural Antimicrobials: A Clean Label Strategy to Improve the Shelf Life and Safety of Reformulated Meat Products

**DOI:** 10.3390/foods11172613

**Published:** 2022-08-29

**Authors:** Norma Angélica Santiesteban-López, Julián Andrés Gómez-Salazar, Eva M. Santos, Paulo C. B. Campagnol, Alfredo Teixeira, José M. Lorenzo, María Elena Sosa-Morales, Rubén Domínguez

**Affiliations:** 1Facultad de Administración, Benemérita Universidad Autónoma de Puebla, Puebla 72592, Mexico; 2Departamento de Alimentos, División de Ciencias de la Vida, Campus Irapuato-Salamanca, Universidad de Guanajuato, Irapuato 36500, Mexico; 3Área Académica de Química, Universidad Autónoma del Estado de Hidalgo, Mineral de la Reforma 42039, Mexico; 4Departmento de Tecnologia e Ciência de Alimentos, Universidade Federal de Santa Maria, Santa Maria 97105-900, Brazil; 5Centro de Investigação de Montanha (CIMO), Instituto Politécnico de Bragança, Campus de Santa Apolónia, 5300-253 Bragança, Portugal; 6Laboratório Associado para a Sustentabilidade e Tecnologia em Regiões de Montanha (SusTEC), Instituto Politécnico de Bragança, Campus de Santa Apolónia, 5300-253 Bragança, Portugal; 7Centro Tecnológico de la Carne de Galicia, Avd. Galicia N° 4, Parque Tecnológico de Galicia, San Cibrao das Viñas, 32900 Ourense, Spain; 8Área de Tecnología de los Alimentos, Facultad de Ciencias de Ourense, Universidad de Vigo, 32004 Ourense, Spain

**Keywords:** pathogenic microorganisms, food safety, healthy meat, functional food, natural additives, natural preservatives

## Abstract

Meat is a nutrient-rich matrix for human consumption. However, it is also a suitable environment for the proliferation of both spoilage and pathogenic microorganisms. The growing demand to develop healthy and nutritious meat products with low fat, low salt and reduced additives and achieving sanitary qualities has led to the replacement of the use of synthetic preservatives with natural-origin compounds. However, the reformulation process that reduces the content of several important ingredients (salt, curing salts, etc.), which inhibit the growth of multiple microorganisms, greatly compromises the stability and safety of meat products, thus posing a great risk to consumer health. To avoid this potential growth of spoiling and/or pathogenic microorganisms, numerous molecules, including organic acids and their salts; plant-derived compounds, such as extracts or essential oils; bacteriocins; and edible coatings are being investigated for their antimicrobial activity. This review presents some important compounds that have great potential to be used as natural antimicrobials in reformulated meat products.

## 1. Introduction

Within human feeding and evolution, meat is an important element playing a key role in diet. Meat and meat products are indispensable for the optimal growth and development of humans in different stages of life because they are sources of protein; vitamins of the B12 complex; and important minerals such as iron, zinc, selenium, and phosphorus [1].

Despite all the benefits that meat provides, epidemiological data have been published about the consumption of red and processed meat being associated with a high risk of developing cardiovascular diseases or obesity as well as an increased risk of certain types of cancer [2]. In recent years, consumers have demanded nutritious and healthy food products, including meat products, for which there are concerns about their high-fat content and high-sodium chloride (NaCl) levels, or the use of some additives such nitrites in processed meats that can produce human health problems [3,4]. To reduce the risk factors, health organizations have recommended reducing portion sizes and limiting the consumption of some meat products [5]. 

Among the compounds and ingredients of meat and meat products, saturated fat is the main component associated with the cardiovascular diseases and cancer [6], which leads to negative implications for consumers’ health and a consequent decrease in meat product consumption. Traditional meat products, such as ground-type meat (patties, burgers, etc.), and those known as emulsified (frankfurter-type sausages and mortadella) have high levels of fat [7,8]. For this reason, different studies have focused on the development of healthy meat products using reformulation and replacing saturated fat with ingredients that provide a health benefit [7,9,10,11,12,13,14,15]. However, replacing or reducing the fat in meat products represents a high technological challenge because the fat provides important sensory and technological advantages (facilitates emulsification, regulates dehydration processes and the final product texture, solubilizes and generates aromas, etc.) [16]. Other reasons for using fat replacers in meat formulation are that water and fat retention, cooking performance, hardness, and cutting and flavor characteristics may be improved [17]. Therefore, the use of multiple fat substitutes based on proteins, carbohydrates, and synthetic compounds were used in meat products reformulation [18]. Additionally, in recent years, research has focused on the addition of vegetable oils to replace animal fat in meat products [16,19,20]. This addition has two goals: to optimize the amounts of lipids and fatty acid profiles as well as to achieve a more convenient composition related to healthy foods. 

Similarly, as with fat, the use of several strategies was proposed to limit and reduce the amount of NaCl added to meat products, including replacing other chloride salts and/or other ingredients with particular flavors/tastes (e.g., mushrooms) [21,22,23]. This is due to the important role of sodium in multiple heart diseases. However, salt is also an important ingredient in meat product manufacturing, not only for its sensory (salt taste) effect or microbial stability but also for its implication in several technological processes, such as dehydration, protein solubilization, enzymatic control, etc. [24]. Thus, the reformulation of meat products and the reduction/replacement of sodium chloride is a challenge to the meat industry and, more concretely, for the control of both spoilage and pathogenic microbial. 

In addition, curing salts (nitrates and nitrites) act as a color stabilizer and are also an efficient inhibitor of *Clostridium botulinum* and the germination of the heat-resistant spores, and other multiple spoilage microorganisms [25]. However, in the recent decade, its use in meat has been controversial, since scientific evidence reported that the consumption of nitrites is associated with a negative impact on human health. This is due to the reactive forms of nitrogen promoting the formation of carcinogenic and mutagenic nitroso compounds, such as N-nitrosamines [25,26]. These compounds are related to several tumors, increasing the risk of leukemia and other damages in cellular structure, proteins, or DNA [27]. In order to overcome this problem, several authors proposed the use of natural extracts and/or essential oils to replace nitrites [28,29]. In fact, the use of vegetable extracts, which naturally contain high nitrate amounts, was proposed to replace this additive [30,31,32]. In this regard, lettuce, celery, chard, spinach, or beet extracts were used [33]. The controlled fermentation of nitrate-rich extracts is also common in pre-converting said nitrates into nitrites and ensuring complete control of the nitrate/nitrite content during meat product processing [34,35,36], since otherwise, there are multiple variables (e.g., the degree of fermentation, the initial nitrate content, pro-/antioxidants in meat product formulation, etc.) that can greatly vary the nitrate/nitrite conversion efficiency and the residual content of nitrites in the final product. Moreover, the plant extracts used as nitrite replacers reduce residual nitrite and N-nitrosamine formation [33]. This allows for the elaboration of meat products without or with reduced nitrite content. Additionally, acids and their salts were proposed to replace nitrites in meat products to limit the risk of growth of *Clostridium botulinum*, *Salmonella*, and *Listeria monocytogenes* [37]. However, their antimicrobial effectiveness should be carefully tested and proven. 

An important point about the addition of non-meat ingredients to replace or reduce fat, salt (NaCl), and/or nitrites in meat products is microbiological safety. The addition of vegetable oils in emulsions leads to the incorporation of water content in meat products, increasing the risk of microbial contamination [38]; the enzymatic and microbial inhibition of sodium chloride salt replacers could vary according to the replacer and the microorganism [39,40]; the diminution of nitrite, as a vital preservative in the meat industry, should also be compensated to limit the growth of pathogenic microbial, such *Clostridium.* Therefore, significant changes in microbiological populations of meat products because of reformulation are expected. Several synthetic preservatives were used for decades to delay microbial activity and to ensure microbial stability, but there are concerns about their toxicity in humans [41]. Thus, natural compounds with preservative activities gained popularity due to their health benefits, environmental issues, and consumer demand [33].

For this reason, to overcome the problems of stability and microbiological safety derived from the strategies of salt, nitrites, and/or fat reduction, different studies have focused on the addition of compounds that prevent the growth of microorganisms, mainly in the addition of multiple compounds as a strategy for developing healthier and new meat products [42,43,44,45,46]. However, consumer demand encourages meat processors to use novel and “clean” or “natural” ingredients as an alternative to commercial chemicals [47,48]. Thus, this review aims to summarize the natural alternative compounds that have the greatest potential to be used as antimicrobials in reformulated healthy meat products.

## 2. Presence of Spoilage and Pathogenic Microorganisms in Meat Products

As in all food industries, the chemical degradative reactions, spoilage, and pathogenic microorganisms are the main cause of economic losses for the meat industry, which also constitute an important consumer health risk [41,49]. In addition to these problems, pathogenic bacteria are a major concern in the meat industry, since foodborne diseases constitute a global health problem [49]. Similarly, microbial spoilage is the main food damage, and 40% of this spoilage occurs at the retail and consumer levels [50]. The main bacterial genera responsible for meat deterioration are *Acinetobacter*, *Alteromonas*, *Aeromonas*, *Brochotrix*, *Flavobacterium*, *Leuconostoc*, *Pseudomonas*, *Moraxela*, Lactic acid bacteria and *Enterobacteriaceae* [51]. Among all of them, the growth of *Lactobacillus* spp., *Staphylococcus* spp., *Carnobacterium* spp., *Leuconostoc* spp., *Lactococcus* spp., *Pseudomonas* spp., *Enterococcus* spp., *Enterobacter* spp., *Acinetobacter* spp., *Moraxella* spp., *Aeromonas* spp., *Psychrobacter* spp., *Serratia* spp., *Enterobacteriaceae*, *Campylobacter* spp., *Escherichia* spp., *Salmonella* spp., *Clostridium* spp., and *Brochothrix* spp., which cause a fast deterioration in meat and food-borne illnesses or produce toxic compounds, such as ammonia, trimethylamine, etc. [3,26,51]. Moreover, the manipulation of meats and meat products during manufacturing, such as in the steps of chopping, slicing, mixing, etc., promoted their contamination, since these actions could increase contact between microorganisms and the product (increases the contact surface) [52]. In this sense, comminuted meat products are highly susceptible to this degradation due to their large interfacial area [53]. 

Another important aspect is that many bacteria strains can bind to environmental or food contact surfaces and form an extracellular matrix (biofilms), which contribute to >80% of human infections [49]. In addition to this problem, biofilms provide microorganisms with additional resistance due to the impossibility of accessing them directly, which prevents antimicrobial compounds, antibiotics, or disinfectant agents from exerting their bactericidal action [54]. This biofilm formation occurs when the bacterial population density reaches a threshold level [55].

Thus, huge difficulties come up because the reformulation of meat products, with the reduction in some vital ingredients such as nitrites, nitrates, or salt, may have a strong influence on the microbial ecosystem and may favor the growth of potentially pathogenic bacteria, resulting in unsafe products [56].

Furthermore, it is important to note that the complex membrane of Gram-negative bacteria (with a protective outer lipopolysaccharide envelope) makes it difficult for some preservatives to perform their antimicrobial action through the mechanisms discussed in the next sections [47]. For example, these lipopolysaccharides present in the Gram-negative membrane repel essential oils, while the bioactive molecules can easily penetrate through the Gram-positive bacterial membrane due to their lipophilic character [49]. Therefore, it is expected that Gram-negative bacteria are less susceptible to the action of natural antimicrobials than Gram-positive bacteria [50], even though several natural antimicrobials exhibit a strong inhibition of both Gram-positive and Gram-negative bacterial growth. Moreover, the interaction of natural molecules with the food components, mainly fat and proteins, reduces their antimicrobial activity. Additionally, the fat of meat products could surround the microorganisms and prevent the action of the natural antimicrobials. Thus, the evaluation of antimicrobial activity in real foods and not only under “in vitro” conditions is vital [50].

*Campylobacter* and *Salmonella* are the most prevalent pathogens in chicken and turkey meat. On other hand, pork presents more *Salmonella* and *Campylobacter*. *Listeria monocytogenes* is the most relevant pathogen for fully cooked and processed ready-to-eat meat products. *L. monocytogenes* survives in processing plants, resulting in post-processing contamination [57]. *Salmonella enterica* ser. *Typhimurium* was the most common serotype associated with laboratory confirmed illness in the past decades.

The effect of fat reduction on the microbiological status of meat products depends on processing variables (such as heat treatment and packaging), storage conditions (temperature and time), and formulation characteristics (meat source, added water, fat level, salt content, and additives) [58]. It was also stated that the addition of 20% water in conjunction with other non-meat ingredients may lead to increased microbial growth in response to greater water activity. For instance, *Staphylococcus aureus*, *Enterobacteriaceae* species, and *Pseudomonas* were found in higher counts for sliced vacuum packaged reduced-fat bologna (9.4% fat) in comparison with bologna with 24.7% fat [59]. 

On the other hand, salt has an inhibitory effect against microorganisms by reducing water activity and the effect of chloride ions. Thus, the reformulation and reduction in salt content in meat products is a problem for maintaining the safety of these products. Furthermore, microbial growth also promotes the formation of biogenic amines, which can react with nitrite derivatives and produce nitrosamines [56].

Similarly, the safety of meat products is also compromised by the trend toward reducing the use of certain additives such as nitrite. This additive possesses strong antimicrobial bioactivity due to the reduction in microbial oxygen uptake and metabolic enzymes and changes the electron transport chain [27]. As aforementioned, nitrites effectively inhibit the germination of spores of *Cl. Botulinum* and other important microbial such as *L. monocytogenes*, *S. aureus*, *Salmonella* spp., *E. coli*, *Cl. Perfringens*, and *B. cereus* [56]. Moreover, *L. monocytogenes* possess a high tolerance to high-salt content and low pH, which represent a huge challenge for the meat industry [27]. Thus, a reduction in nitrites can promote the growth of this pathogen and cause foodborne problems. 

Therefore, due to the safety of meat products potentially becoming out of control, it is clear that it is vital to find alternatives to these controversial ingredients since we must ensure that the products marketed are perfectly safe and do not pose any risk to human health. In fact, outbreaks related to foodborne infections are a major concern among consumers, the meat industry, and health organizations and authorities [60], and among all, about half of outbreaks are linked to contaminated meats [61]. That is why a multitude of natural compounds with antimicrobial activity have been proposed to supply the antimicrobial functions of these ingredients in meat products.

## 3. Natural Antimicrobial Agents and Their Effect on Meat Products

### 3.1. Organic Acids

Organic acids could be a suitable option as natural antimicrobials in reformulated meat products. They have shown clear efficiency; however, sensory changes (color and flavor) might be taken into consideration. The most common compounds are acetic, citric, lactic, propionic, malic, succinic, and tartaric acids (Figure 1). They are excellent antimicrobials against bacteria and have been considered (organic acids and their salts) as the most popular preservatives in meats [61,62] (Table 1), with diverse advantages such as GRAS (generally regarded as safe) recognition, no limited acceptable daily intake, low cost, and being easy to manipulate. However, it is important to use these acids according to good manufacturing practices to avoid the development of *Salmonella* strains, which are resistant to acidic conditions [63]. 

The natural origin of organic acids makes their use simple and cheap. For example, citric acid can easily be extracted from several citrus juices and acidic fruits, while benzoic acid is present in multiple vegetables (cranberries, cloudberries, etc.) [62]. Moreover, the metabolism of lactic acid bacteria also produces organic acids via carbohydrate fermentation. 

Among all organic acids, lactic, acetic, and citric acids are considered the most interesting, since their inclusion as preservatives exerts a clear antimicrobial effect against several pathogenic and spoilage bacteria, while as they occur naturally in foods, improve the taste (sensory quality) of multiple foods, and have various techno-functional functions [62].

Organic acids as well as their salts have a potent inhibitory effect against multiple bacteria (Gram-positive and Gram-negative bacteria) and an important antifungal spectrum. It is important to explain that the isoelectric point (point of zero charge or in which the molecule is not dissociated) is vital for their antimicrobial activity, and the pH regulates the dissociate/undissociated equilibrium, which affects the ability of organic acids to penetrate in the bacterial cells. In fact, the dissociation constant of some organic acids proposed as antimicrobials is normally <5 (acetic acid 4.76; succinic acid 4.21; lactic acid 3.86; citric acid 3.13; malic acid 3.40) [64]. The slightly acidic pH in meat products (pH_out_) allows organic acids, due to their weak dissociation constants, to maintain a large proportion of their undissociated form. Thus, the uncharged form of organic acids has the ability to freely move across the bacterial membranes, and then, they dissociate inside the bacterial cytoplasm [65] (pH_in_ > isoelectric point), which produces the toxic anion and inhibitory effects, as well as a decrease in the intracellular pH. Moreover, the addition of organic acids and salts produces a decrease in the pH of the food, which produces a direct inhibition of microbial growth, and this inhibitory effect increases at low pH. Organic acids promote the disruption of the proton motive force and create an unfavorable environment for microorganisms [66]. The intracellular pH variations also induce cellular damages, which include enzymes and protein modifications, and DNA changes. This inhibits the metabolism of the microbial and causes cell death [62]. In a more simple way, other authors concluded that these antimicrobials induce cytoplasmic acidification and lead to an imbalance of energy, and the accumulation of toxic levels of acid anions, which result in inhibition or death of microbial [61]. 

Additionally, the cell membrane disruptions are related to the proton motive force, which results in energy depletion and the inhibition of essential metabolism reactions. Therefore, taking into account that undissociated acids can cross the microbial membrane (as aforementioned), the organic acids possess stronger antimicrobial activity than inorganic acids (high dissociation rate), which suggests the application of these acids (organics) in food preservation. 

Although this general statement is valid for roughly explaining the antimicrobial actions of organic acids, it should be noted that there is no single mechanism and that their preservative activity is due to various and specific methods of action [62]. For example, lactic acid produces destabilization in membrane molecular interactions in Gram-positive bacteria, which causes pores in the membrane and cell death. Additionally, both lactic and acetic acids produce changes in the metabolic and anabolic processes, while disturbing the transmembrane proton force and denaturing proteins and DNA. Finally, citric and malic acids induce metabolic changes, since they chelate essential ions and destabilize the cellular membrane, which affects microbial growth. 

Generally speaking, the amount of organic acids (lactic, acetic, and citric) applied to the foods ranged between 0.1 and 0.4%, while in the case of sorbic and benzoic acids, this amount was between 1000 and 2000 ppm and, for propionic acid, was 2500 ppm [62]. The limits for sorbic and benzoic acid are due to some studies reporting toxicity (mutagenic, genotoxic, and carcinogenic) of these compounds and their salts. 

In any case, despite the promising results that some of them show, until now, there have been very few practical applications of organic acids and their salts in the reformulation of meat products, but its application is frequently used for the decontamination of meat (for example, decontamination of carcasses in slaughterhouses). In addition, in order to avoid toxicological problems and allergic reactions, both the effects on the meat and the absence of interactions with it must be checked. 

Moreover, acetic acid is a monocarboxylic acid with a sharp smell and taste, which restricts its use in food. It is highly soluble in water and can be found in marinated products such as sausages and pork legs [63]. Additionally, the sensitivity of *Salmonella* to the effect of acetic acid is greatly impaired by the physiological state of the bacteria studied. The relative sensitivity of the six *Salmonella* serovars to the acetic acid has been shown in the following order: *Bareilly*, *Typhimurium*, *Montevideo*, *Poona N*, *Mbandaka*, and *Stanle* [67]. 

Sodium lactate (3%), a salt derivative from lactic acid, was used as an antimicrobial agent and compared with trisodium phosphate and potassium sorbate for low-fat Chinese-style sausages. Better physicochemical properties and lower microbiological counts (about 0.5–1 log CFU/g lower than control samples for total plate count and total anaerobic count) were achieved for sodium lactate during the storage of the sausages (18% fat) for 12 weeks at 4 °C [68]. 

Sodium lactate also effectively inhibits the growth of different pathogenic microbial on inoculated cooked ham (2.9% fat). The use of 1 or 2% of lactate produces a delay in the *L. monocytogenes*, *E. coli*, and *Salmonella* spp. at multiple storage temperatures [69]. In this case, the average growth rates of *L. monocytogenes* in control samples were 0.256 log CFU/day, while in samples reformulated with sodium lactate, this growth rate decreased to 0.158 (1% of sodium lactate) and 0.104 log CFU/day (2% sodium lactate) in samples stored at 4 °C, which represent a 35% and 60% reduction in average growth rate, respectively. Similarly, for samples stored at 8 °C, the reduction in this pathogen was 16% and 41% (for samples treated with 1% and 2%, respectively), while for samples stored at 10 °C the reduction was 16% and 34%. Similarly, the average growth rate of *E. coli* at 8 °C was reduced by the addition of 1% (33% reduction) and 2% (50% reduction) of sodium lactate, while for *Salmonella* spp., the growth rate was reduced by 55% and 79% in samples stored at 10 °C and treated with 1% and 2% of sodium lactate, respectively [69]. It is also important to highlight that these authors reported a significant reduction in average growth in all stored temperatures, but the sodium lactate inhibition was more effective at low temperatures [69]. 

Likewise, ground beef meat treated with 3% sodium lactate presented, after 21 days of storage at 4 °C, significantly lower aerobic plate (9.69 vs. 6.73 log CFU/g for control and samples with 3% of sodium lactate, respectively), psychrotrophic (9.98 vs. 7.79 log CFU/g for control and samples with 3% of sodium lactate, respectively), *Enterobacteriaceae* (7.39 vs. 5.19 log CFU/g for control and samples with 3% of sodium lactate, respectively)*,* and lactic acid bacteria counts (8.36 vs. 7.28 log CFU/g for control and samples with 3% of sodium lactate, respectively), which demonstrated its effectiveness in increasing the ground meat shelf-life (from 8 to 15 days) [70]. The application of sodium lactate (1, 2, or 3%) was also effective in delaying the growth of aerobic plate counts (about 1.5–2 log reduction) in ground pork meat [71].

In another study, the authors used sodium lactate (3–6%), sodium acetate, or sodium diacetate (0.25–0.5%) to control the growth of *L. monocytogenes* in inoculated frankfurter-type sausages [72]. The application of 6% of sodium lactate (between 2.4 and 6.4 log reduction) or 0.5% of sodium diacetate (between 2.4 and 6.6 log reduction) produces a clear bacteriostatic effect at all storage times (120 days), while 3% of sodium lactate inhibited the pathogen growth during the first 70 days (between 2.4 and 4.1 log reduction in comparison with the control). The same findings were obtained in frankfurters inoculated with *L. monocytogenes* treated with sodium lactate (1.8%) (~1.5 log reduction) and sodium diacetate (0.25%) (~2 log reduction) or their combination [73], in which the combination of both salts completely inhibited the *L. monocytogenes* growth (about 2 log CFU/cm^2^ constant during 40 storage days), while also reducing the total microbial counts (between 1 and 5 log reduction, depending on the antimicrobial), and yeast and molds counts. These authors prove that the same salt combination (1.8% sodium lactate with 0.25% sodium diacetate) was also effective for controlling the growth of both, pathogen *L. monocytogenes* (~5.5 log reduction) and spoilage total microbial populations (~5 log reduction) during the storage of pork bologna sausage under refrigeration temperatures (4 °C) [74]. 

The addition of potassium lactate (2%) to three different ready-to-eat meat products (poultry Bologna, pork ham, and roast beef), also showed a strong reduction in *L monocytogenes* and *Salmonella* spp. counts during storage [75]. These authors showed that the addition of potassium lactate extended the lag phase of both types of microorganism, and then, during storage at 4 °C, treated samples had a ~1.5 log reduction in roast beef, and about a 1 log reduction in poultry deli loaves and cured pork hams in comparison with untreated samples [75].

The effects of nine organic acids (30 g/L) and peroxyacetic acid (2 g/L) as an antimicrobial treatment in beef trimmings were tested [76]. In general, most organic acids were effective at reducing bacterial populations, but caprylic acid (most effective) produces a strong reduction in coliforms (4.78 log reduction), *E. coli* O157:H7 (4.73 log reduction), and aerobic plate count bacteria (2.48 log reduction), and pyruvic acid also showed important antimicrobial activity against these microbial (1.84, 1.68, and 1.08 log CFU/g, respectively) [76]. 

As a general conclusion and taking into account the aforementioned, the application of organic acids and their salt is a promising strategy to extend the shelf-life of meat products as well as to ensure their safety due to the strong inhibitory effect that they possess against pathogenic and spoilage microbial. However, it is also important to highlight that the insufficient preservative effect of organic acids and their salts could be related to possible interferences with other antimicrobials or with compounds found in the meat [62]. 

### 3.2. Plant-Based Compounds

Recently, plant derivatives were proposed as natural preservatives in the meat industry [3,77]. Many plant compounds, mainly polyphenols, have demonstrated their antimicrobial activity against several pathogens, both in vitro and in food models, and thus, they may be employed as food preservatives in reformulated meat products in order to improve their microbial safety [47,78]. A common feature of these phenolic compounds is the presence of one or more aromatic rings in their structure, and one or more -OH groups, which are essential for antibacterial properties. Common phenolic compounds include phenolic acids (ferulic, p-coumaric, ellagic, rosmarinic, caffeic, and gallic acids), phenolic diterpenes (carnosic and carnosol acids), flavonoids (quercetin, catechin, rutin, and kaempferol) (Figure 2), and volatile essential oils (menthol, carvacrol, thymol, eugenol, cymene, pinene, terpinene, cinnamaldehyde, camphor, and linalool) [33,53,79]. Thus, taking into account their properties (lipophilic or hydrophilic character) and composition, plant-based compounds could be subdivided into extracts (normally hydrophilic, obtained by aqueous or hydroalcoholic extractions) and essential oils (lipophilic character, obtained using distillation procedures or extractions with organic solvents). Moreover, both plant extracts and essential oils were used as different strategies to prolong the shelf-life of meat and meat products, and they possess strong antimicrobial and antifungal activity; thus, they are a natural and good alternative to synthetic preservatives [3]. The main antimicrobial compounds in plant extracts and essential oils and the microorganisms they inhibit are summarized in Table 2.

The molecular mechanism of the biological activity of polyphenols is not yet well understood. In addition, it is usually necessary to add a high amount of polyphenol-rich extracts to obtain an effective antibacterial activity. In this way, research on the assessment and optimization of synergies using natural plant-based preservatives has led to knowledge on the production of meat products without synthetic preservatives during more extended storage periods [53].

The use of these derivatives as natural antibacterial agents in reformulated meat products can be useful in improving product quality and helps the industry meet the demand for healthier and safer products than synthetic ones. Nonetheless, it is necessary to extract and separate secondary plant substances, to confirm their antibacterial effects, and to consider potential applications for the development of healthier and more stable meat products [42]. Additionally, the use of plant extracts instead of the direct addition of plant powders has multiple advantages. The bioactive molecules are concentrated form in these extracts; thus, they can exert their potent antimicrobial activity at lower concentrations [33], increase the stability of the antimicrobials, reduce transport and storage costs, homogenize their composition, and facilitate their application in meat products. Nevertheless, the polyphenol mixtures are more effective than single compounds; thus, the application of plant extracts is more convenient rather than individual compounds [26]. 

Due to the high content of phenolic compounds, fruits and other plant materials are a potential source of antimicrobial-active molecules. However, it is necessary to optimize the amount added so as not to impair the sensory quality and, thus, affect acceptance of the product. Consumers may view these changes as negative. Depending on the product, these flavors can be considered negative or positive [80]. This is of special importance in essential oils since they contain high amounts of terpenes and other compounds with a very low odor threshold and with characteristic aroma. Anyway, plant-derived extracts and essential oils have been recognized as natural antimicrobials and proven in diverse meat matrices by inhibiting or reducing microbial degradation [45,79]. Some spices are routinely used in meat products and are likely to act on them by inhibiting bacterial growth, depending on the amounts used in the formulation. 

The extracts obtained from pomegranate peel have several phenol compounds, including phenolic acids (caffeic, gallic, ellagic, or p-coumaric acids), flavonols (quercetin), flavonols, and anthocyanins, which act as effective antimicrobials against food borne pathogen bacteria, such as *Salmonella* spp, *Listeria monocytogenes*, *Bacillus subtilis*, *Escherichia coli*, *Staphylococcus aureus*, or *Pseudomonas aeruginosa*, among others [81]. However, it is important to note that the phenol’s antimicrobial activity is dose-dependent, and it has been proposed that, at low concentrations, these compounds inhibit microbial enzymes while, at high concentrations, they produce protein denaturation. 

Red fruit extracts (plum, red grapes, and elderberries) also presented a high microbial inhibition against pathogens *B. cereus*, *S. aureus*, and *E. coli*, which is related to the high anthocyanins contents of the extracts. Although these extracts presented a high antimicrobial activity against pathogens, it is important to highlight that they stimulated the growth of the probiotic bacteria [82].

Grape seed extracts had important phenolic content, which includes hydroquinone; pyrocatechol; caffeic, ferulic, ellagic, ρ-coumaric, protocatechuic, caftaric, ρ-hydroxybenzoic, syringic, and gallic acids; resveratrol; flavan-3-ols; catechin; epicatechin; quercetin-3-O-rhamnoside; and procyanidins. This extract was effective against *B. cereus*, *B. coagulans*, *B. subtilis*, *S. aureus*, *E. coli*, *B. thermosphacta*, and *P. aeruginosa* [47].

Green tea extracts are also a potent antimicrobial, characterized by high amounts of catechins (epicatechin, epigallocatechin, epicatechin gallate, and epigallocatechin gallate), which have an important inhibitory effect on multiple microbial, including *Campylobacter jejuni*, *Staphylococcus aureus*, *Escherichia coli*, *Listeria monocytogenes*, *Salmonella Typhimurium*, *Vibrio parahaemolyticus*, *Bacillus cereus*, *Pleisomonas shigelloides*, *Clostridium perfringens*, and *Pseudomonas fluorescens* [41].

Various studies reported high preservative properties of olive leaf extract. Its antimicrobial activity is related to the high content of oleuropein, which is the main phenolic component, and to flavonoids and phenolic compounds, including oleuroside, demethyloleuropein, ligstroside, verbascoside, and non-glycosidic secoiridoids [3]. Moreover, the amount of hydroxytyrosol; tyrosol; caffeic, ρ-coumaric, and vanillic acids; vanillin; luteolin; diosmetin; rutin; verbascoside; luteolin-7-glucoside; apigenin-7-glucoside; diosmetin-7-glucoside; rhamnetin; isoquercitrin; kaempferol; kaempferitrin; saponins; triterpenoids; tannins; anthraquinones; alkaloids; and terpenoids was also reported to be important [33,47]. Regarding its antimicrobial activity, the use of olive leaf extract completely inhibits the growth of *Salmonella enteritidis* and *Listeria monocytogenes*, and the authors attributed this inhibitory effect to the presence of oleuropein and verbascoside compounds [83]. Other studies also reported that olive extracts have antimicrobial effects against *S. aureus*, *C. jejuni*, *S. enterica*, *L. monocytogenes*, and *P. aeruginosa* [47].

*The Lamiaceae* family contains important plants with bioactive and antimicrobial compounds. For example, rosemary is rich in phenolic acids such as rosmarinic, caffeic, ursolic, betulinic, and carnosic acids and in terpenes such as camphor and carnosol; sage extracts contain mainly rosmarinic acid, carnosic acid, and carnosol; thyme presents important amounts of romarinic, caffeic, p-hydroxybenzoic, p-cumaric, ferulic, protocatechuic, syringic, and chlorogenic acids as well as quercetin, luteolin, and apigenin; and Saruteja has important contents of caffeic, isoferulic, and rosmarinic acid as well as naringenin and apigenin [3,84]. 

Citrus extracts also have a bioactivity effect due to them preventing cell wall synthesis, influencing the metabolism of nucleic acids, and promoting cell lysis. These effects are related to the presence of flavonoids, phenolic acids, and carotenoids. In this sense, the orange peel extract possessed a strong bactericidal activity against *E. coli*, *B. subtilis*, *S. aureus*, and *Xanthomonas citri* [85].

The main active molecules identified in roselle calyx extract were gallic acid, catechin, epicatechin, chlorogenic acid, protocatechuic acid, and hydroxycinnamic acids. These compounds exert a clear antimicrobial effect against *E. coli*, *Salmonella enterica*, *Salmonella typhimurium*, *S. aureus*, *L. monocytogenes*, and *B. cereus* [53]. 

Onion and garlic extracts have organosulfur compounds, which include allylsulfide, diallilsusfide, alliin, propylsulfide, s-methyl-cysteine sulfoxide, S-methyl methanethiosulfonate, and cycloallicin, but also presented important amounts of catechins, gallic acid and its derivatives, and kaempferol derivatives. All of the aforementioned compounds have demonstrated potent antimicrobial activity against bacteria and fungi [3,86]. In fact, allium extract compounds demonstrated a bioactivity effect against *L. monocytogenes*, *S. enteritidis*, *E. coli*, *P. hauseri*, and *E. faecalis* [86]. This antimicrobial activity could be attributed to the thiosulfinates and their S-(O)-S groups that react with –SH groups of cellular proteins and affect lipid synthesis, which influences the structure of the bilayer membrane [86]. 

The antimicrobial activity of the phenolic compounds could be summarized as follows [26,33,50,53,81]: the hydroxyl groups produce leakage of cellular content and dissipation of cellular energy (ATP), which leads to cell death. Additionally, the increase in proton concentration produces a pH drop and decreases the internal cellular pH. Phenolic acids act in the membrane integrity since these compounds cause membrane disruption (e.g., effective against *Listeria monocytogenes*) and coagulation of cell contents. In the case of anthocyanins, there is a strong *E. coli* inhibitor, which increases the Gibbs free energy of adhesion to cells, which discourages the attachment of bacteria to these cells. Tannins form a complex with bacterial polysaccharides and metals necessary for microbial metabolism and can precipitate membrane proteins, which produce enzymatic inhibition, oxidative phosphorylation, lysis, and microbial death. Flavonoids act in membrane proteins and reduce the membrane fluidity, while also can produce modifications to energy metabolism, DNA, protein, and RNA syntheses. Finally, other plant extract compounds such as alkaloids exert their antimicrobial action by intercalation with DNA and inhibition of the synthesis of nucleic acids, which prevent microbial proliferation. It is also important to highlight that, although hydroxyl groups are the main ones responsible for the antimicrobial activity of polyphenols, their number and their position exert a clear influence on this activity [50,53]. Therefore, the greater the number of hydroxyl groups, the greater its antimicrobial activity. In addition, the molecular position also affects its effectiveness in inhibiting microbial growth, and it is speculated that this is because this position is relevant in the delocalization of electrons from the cytoplasmic membrane and, therefore, in the bactericidal activity.

With all this in mind, several studies proposed the application of these extracts to increase the shelf-life of meat products and/or inhibit the growth of pathogens.

The olive and apple extracts were used (at 1 and 3%) to control the growth of pathogenic *E. coli* O157:H7 in beef patties [87]. The authors observed a strong bactericidal activity of olive extract (3%) and a reduction in this pathogen to below the detection limit, and a significant reduction in the pathogen survivors was achieved with the addition of 1% (0.7 log reduction) or 3% (1 log reduction) of apple extract. This demonstrated that both extracts are susceptible to use in the reformulation of meat products in order to control the growth of this pathogen microbial [87].

Extracts obtained from green tea, stinging nettle, and olive leaves and applied at 500 ppm produced a clear antimicrobial (total viable count) and antifungal (yeast and molds) activity during frankfurter-type sausage storage [41]. This study concludes that all extracts inhibit microbial growth (>1 log reduction with green tea and stinging nettle extracts), which results in a shelf-life extension of this meat product. In a more recent study, taking into account the promising results, the same researchers proposed the use of different amounts of nisin, ε-polylysine, or chitosan in combination with a mixed extract (green tea, stinging nettle, and olive leave extracts) in nitrite-free frankfurter sausages [25]. The authors reported that combinations of 0.2% ε-polylysine or 1% chitosan with mixed extract effectively inhibit the total viable count (between 0.5 and 1 log reduction, depending on the storage day), and the growth of yeasts and molds (>1.5 log reduction). This demonstrates that both combinations are effective at producing nitrite-free sausages and, at the same time, extending their self-life (30% shelf-life in comparison with control samples). The authors attributed the chitosan efficiency to the interaction between positively charged residues in chitosan with the negatively charged molecules of microbial membranes, chelating metals, and the interaction and inhibition of protein synthesis. Similarly, the polylysine interacts with microbial membrane and enzyme activity, while extracts inhibit microbial growth through the mechanisms before discussed [25].

**Table 2 foods-11-02613-t002:** Main bioactive compounds and antimicrobial activity spectrum of plant-derived extracts and essential oils.

Plant Material	Bioactive Compounds	Antimicrobial Activity Spectrum	Ref.
Pomegranate peel (Extract)	Phenolic acids (caffeic, gallic, ellagic, or p-coumaric acids), flavonols (quercetin), flavonols, and anthocyanins	*Salmonella* spp, *L. monocytogenes*, *Bacillus subtilis*, *Escherichia coli*, *S. aureus*, or *P. aeruginosa*.	[81]
Grape seed (Extract)	Hydroquinone, pyrocatechol, caffeic, ferulic, ellagic, ρ-coumaric, protocatechuic, caftaric, ρ-hydroxybenzoic, and syringic and gallic acids, resveratrol, flavan-3-ols, catechin, epicatechin, quercetin-3-O-rhamnoside, and procyanidins	*B. cereus*, *B. coagulans*, *B. subtilis*, *S. aureus*, *E. coli*, *B. thermosphacta*, and *P. aeruginosa*.	[47]
Green tea (Extract)	Epicatechin, epigallocatechin, epicatechin gallate, and epigallocatechin gallate	*C. jejuni*, *S. aureus*, *E. coli*, *L. monocytogenes*, *S. Typhimurium*, *V. parahaemolyticus*, *B. cereus*, *P. shigelloides*, *Cl. perfringens*, and *P. fluorescens*.	[41]
Olive leaf (Extract)	Oleuropein, oleuroside, demethyloleuropein, ligstroside, verbascoside, non-glycosidic secoiridoids, hydroxytyrosol, tyrosol, caffeic, ρ-coumaric, and vanillic acids, vanillin, luteolin, diosmetin, rutin, verbascoside, luteolin-7-glucoside, apigenin-7-glucoside, diosmetin-7-glucoside, rhamnetin, isoquercitrin, kaempferol, kaempferitrin, saponins, triterpenoids, tannins, anthraquinones, alkaloids, and terpenoids	*S. aureus*, *C. jejuni*, *S. enterica*, *L. monocytogenes*, *P. aeruginosa*, *S. enteritidis*, and *L. monocytogenes*.	[3,33,47,83]
Roselle calyx (Extract)	Gallic acid, catechin, epicatechin, chlorogenic acid, protocatechuic acid, and hydroxycinnamic acids	*E. coli*, *S. enterica*, *S. typhimurium*, *S. aureus*, *L. monocytogenes*, and *B. cereus*.	[53]
Onion and garlic (Extracts)	Allylsulfide, diallilsusfide, alliin, propylsulfide, s-methyl-cysteine sulfoxide, S-methyl methanethiosulfonate, cycloallicin, catechins, gallic acid and its derivatives, and kaempferol derivatives	*L. monocytogenes*, *S. enteritidis*, *E. coli*, *P. hauseri*, and *E. faecalis*.	[3,86]
Sage (Essential oil)	α-Thujone, camphor, and eucalyptol, viridiflorol, epirosmanol, β-thujone, borneol, bornyl acetate, trans-caryophyllene, and α-humulene	*S. aureus*, *E. coli*, *B. subtilis*, *P. aeruginosa*, and *A. niger*	[88,89]
Thyme (Essential oil)	Thymol, carvacrol, p-cymene, linalool, γ-terpinene, terpinen-4-ol, α-terpinene, β-myrcene, camphene, geraniol, borneol, α-terpineol, camphor, limonene, β-pinene, trans-caryophyllene, borneol, α-himachalene, γ-elemene, and sabinene	*L. monocytogenes*, *S. aureus*, *E. coli*, *S. typhimurium*, *B. licheniformis*, *L. innocua*, *P. fluorescens*, *P. vulgaris*, and *P. putida*.	[49,84,90,91]
Cove (Essential oil)	Eugenol, eugenyl acetate, β-caryophyllene, 2-methoxy-4-(2-propenyl)-phenol acetate, α-humulene, and α-caryophyllene	*L. monocytogenes*, *S. aureus*, *E. coli*, *S Typhimurium*, *S. enterica*, and *C. jejuni*.	[47,92]
Lemongrass (Essential oil)	Citral, geranial, neral, myrcene, limonene, cosmene, o-cimene, α-terpinolene, verbenol, citronellal, linalool, cis-carveol, nerol, atrimesol, carveol, geranyl acetate, and caryophyllene	*L. monocytogenes*, *Yersinia*, *E. coli*, *Staphylococcus* spp., *S. Typhimurium*, *L. plantarum*, *P. aeruginosa*, *B. cereus*, *B. subtilis*, *E. faecalis*, and *E. aerogenes*.	[93]

The counts of total psychrotrophic and aerobic mesophilic microorganisms from fresh chicken sausages treated with 1% lemongrass extract were significantly lower than those in the control samples (~1 log reduction for both types of microbial) [94]. Similarly, the shelf-life of cooked and shredded chicken breast was increased with the addition of lemongrass extract, and the presence of *Staphylococcus* spp., *Salmonella* spp., and *Coliforms* was not detected [95]. The antimicrobial activity of this extract is related to the active compounds citral, neral, and geraniol [53]. 

The application of roselle calyx extract was proposed to increase the self-life of steaks [96] and ground beef meat [97]. In these studies, the in vitro analysis showed that this extract inhibited the growth of several pathogens, including *E. coli*, *Salmonella enterica*, *Salmonella typhimurium*, *S. aureus*, *L. monocytogenes*, and *B. cereus*, while their application in meat produce a clear antimicrobial effect against psychrophiles and mesophiles in steaks (about 1 and 3 log reduction, except on the last day of storage) [96] and against *E. coli* (5.7 and 3.8 log reduction for ethanolic and aqueous extracts, respectively), *Salmonella typhimurium* (6.0 and 4.6 log reduction for ethanolic and aqueous extracts, respectively), *S. aureus* (6.1 and 4.8 log reduction for ethanolic and aqueous extracts, respectively), *L. monocytogenes* (6.0 and 4.1 log reduction for ethanolic and aqueous extracts, respectively), and *B. cereus* (6.3 and 4.4 log reduction for ethanolic and aqueous extracts, respectively) in ground beef meat [97]. 

Beetroot is another promising vegetable used to obtain valuable extracts. In a study conducted this year, the authors proposed the use of beetroot extract and proved their antibacterial activity against *L. monocytogenes* in cooked pork [98]. The results indicated that this extract produces apoptosis-like death in *L. monocytogenes* and has a strong potential function as a natural antimicrobial, more specifically against *L. monocytogenes* (0.27 and 0.84 log reduction using 1 MIC and 2 MIC concentrations, respectively). These authors justified the bioactivity of beetroot extract due to the decrease in intracellular ATP, which produces a dissipation of proton motive force components, with membrane depolarization, ROS depletion, and DNA fragmentation in *L. monocytogenes* [98]. 

In another very recent study, the authors concluded that the application of strawberry tree and pomegranate peel extracts showed a clear antilisterial activity when they were incorporated into a dry-cured ham-based model [78]. In this study, the authors reported that control samples showed >7 log CFU/mL, while in samples treated with strawberry tree extract, the L. monocytogenes counts were 4.14 and 5.19 log CFU/mL and those treated with pomegranate peel extract were 4.26 and 5.27 log CFU/mL.

An extract from *Amaranthus tricolor* was also tested as an antimicrobial against *S. aureus* and for their potential application in preventing foodborne illnesses in cooked meat [99]. This extract presented bioactivity against this pathogen (between 0.5 and 1 log reduction) and, according to the authors, possesses latent energy for becoming an excellent biopreservative used to maintain the safety of cooked pork. The mechanisms that explain the antimicrobial activity were due to the extract (rich in alkaloids, polyphenols, terpenoids, and saponins) producing membrane depolarization; a decrease in pH; a leakage of intracellular components; DNA cleavage; cell deformation; and finally, the destruction of membrane structure and cell disintegration (Figure 3) [99]. 

Higginbotham et al. [100] used extracts from Jamaica as an additive in sausages to inhibit the reproduction of *Listeria monocytogenes* and *Staphylococcus aureus* resistant to methicillin and concluded that it is highly effective for the inhibition of these species bacterial and that, at higher concentration and time the immersion, the better the antimicrobial effect. In inoculated hot dogs immersed for 60 min in 240 mg/mL of hibiscus extract, the authors observed a complete reduction in *S. aureus* (from 5.56 log CFU/g to non-detectable), while producing a 4.05 log reduction in *L. monocytogenes*, both after 24 h.

On the other hand, not only the chemical composition but also the structure are important for the antimicrobial effect of natural compounds, since their lipophilic properties exert a vital role in their antibacterial activity. In this regard, the antimicrobial activity of plant-derived compounds increases with the elevation of their lipophilic character due to the increase in the potential interaction with lipidic compounds of microbial membrane and the ability to penetrate this membrane [50]. Several lipophilic-derived substances can be obtained from plants, including essential oils, non-essential oils, and resins. Resins (which include oleoresins, balsams, varnish, and lacquer resins) are secondary plant metabolites constituted by a complex mixture of essential oils, phenolic compounds, and fatty substances (also known as non-essential oils) [101]. Among all types of resins, high contents of organic acids such as benzoic or cinnamic acids can be found in balsams [102]. These balsams include conifer, leguminous, styrax, and myrrh, among others [101]. Thus, multiple plant genera such as Myroxylon and Styrax produce resins with high amounts of some organic acids with important antibacterial properties [103].

In addition to organic acids, the major chemical components of *Styrax officinalis* (used in traditional medicine) are (E)-2-hexenal, octanol, and geraniol. The *S. officinalis* oleoresin is also known as “Storax”, and it was used to cure respiratory infectious diseases and microbial infections [104]. *S. officinalis* extracts (aqueous and organic extracts) showed antimicrobial activity against *S. aureus*, *E. coli*, and *P. aeruginosa* [105], and these authors conclude that these extracts can be used as natural food preservatives. Similarly, other studies reported important bioactive compounds, which include egonol, multiple benzofurans, phenolic acids, flavonoids, saponins, fatty aldehydes, terpenes, and terpenoids; organic acids (succinic and lactic acid); and several lipid-derived compounds [104]. Due to their composition, various authors reported that organic and aqueous extracts obtained from leaves and fruits have strong antimicrobial against *S. aureus*, *P. aeruginosa*, and *E. coli* [105,106], with a similar inhibitory effects that with potent broad-spectrum antibiotics.

Chio resin (also known as Chios mastic gum) is another important resin obtained from *Pistacia lentiscus* var. *chia* and is a phytotherapeutic remedy traditionally used to treat gastrointestinal disorders since it possesses antibacterial properties due to its composition rich in triterpenes [107]. This resin has about 3% of essential oil, in which α-pinene, β-myrcene, p-cymene, terpinen-4-ol, and β-pinene are the major components (~90%) [108,109]. The main use is for inhibition of *Helicobacter pylori* and oral pathogens, but it also seems to be effective against several food-borne microorganisms, including *S. aureus*, *Lactobacillus plantarum*, *Pseudomonas fragi*, and *Salmonella enteritidis* [107]. Non-essential oils extracted from *Pistacia lentiscus* L. also possess potent antimicrobial properties (in vitro) against *S. aureus* [110]. 

However, very few studies reported the use of resins as a food ingredient. For example, myroxylon-derived flavoring ingredients are not known to be used as a food, although some of them were considered as GRAS (e.g., FEMA2117 or FEMA 3069) [111]. 

In contrast to resins, several essential oils (cinnamon, sage, rosemary, oregano, basil, thyme, clove, nutmeg, menthol coriander, ginger, and lavender essential oils) and their active compounds (vanillin, carvacrol, citral, cinnamaldehyde, linalool, carvone, eugenol, thymol, and limonene) have been generally considered as GRAS [79,112], and some of them were approved as food additives in the USA., such as lemon balm, basil, clove, vanilla, thyme, and coriander [49]. Thus, essential oils were proposed and approved for use in the food industry. Moreover, it is expected that essential oils, with a marked lipophilic character, present high antimicrobial activity [79]. The main constituents of essential oils are terpenes, phenylpropanoids, aldehydes, esters, alcohols, and ketones, but normally, the most abundant groups are terpenes and their oxygenated derivatives (Figure 4) [51,79,112]. Thus, although the bioactivity of essential oils is highly dependent on their components [113], the essential oils present important antimicrobial and antifungal molecules [48]. For example, the main bioactive compounds in oregano are thymol and carvacrol (about 80%), but ρ-cymene, linalool, thujene, myrcene, caryophyllene, and γ-terpinene were also observed in this essential oil. High amounts of 1,8-cineole, α-pinene, limonene, and camphor were reported in rosemary essential oils, and less camphene, borneol, bornyl acetate, and α-terpineol were reported. Similarly, from Satureja, borneol, carvacrol, linalool, thymol, γ-terpinene, and ρ-cymene can be isolated as the main compounds of essential oil, and the antimicrobial activity of pepper essential oil is related to its terpenoids, in particular with the high amounts of caryophyllene, limonene, α-terpinene, and α-pinene [3]. 

In a study, the authors identified twenty-six different compounds in Satureja essential oil, but the main components were thymol, p-cymene, linalool, and carvacrol [114].

In sage essential oil obtained by supercritical fluid extraction, several compounds were identified, in which the most important bioactive molecules were oxygenated monoterpenes (α-thujone, camphor, and eucalyptol), oxygenated sesquiterpenes (viridiflorol), and diterpene polyphenols (epirosmanol), followed by β-thujone, borneol, bornyl acetate, trans-caryophyllene, and α-humulene [88]. In another recent study, the authors identified ciridiflorol as the main compound and important amounts of verticiol, camphor, borneol, α-thujone, β-thujone, and eucalyptol [89]. Additionally, this essential oil has bioactivity against *S. aureus*, *E. coli*, *B. subtilis*, *P. aeruginosa*, and *Aspergillus niger* [89]. However, sage, like the other essential oils, has a wide variability depending on the sensitivity of microorganisms. 

Thyme essential oil is another natural antimicrobial that possesses a clear inhibition against both spoilage and pathogenic microbial [49]. More than sixty active molecules were described by multiple authors, but the most common were thymol carvacrol, p-cymene, and linalool, but other compounds, such as γ-terpinene, terpinen-4-ol, α-terpinene, β-myrcene, camphene, geraniol, borneol, α-terpineol, camphor, limonene, β-pinene, trans-caryophyllene, borneol, α-himachalene, γ-elemene, and sabinene hydrate were also identified in medium–high amounts [49,84,90]. These volatile compounds presented a strong inhibition effect against *L. monocytogenes*, *S. aureus*, *E. coli*, and *S. typhimurium*, while having no antimicrobial activity against *P. aeruginosa* [49]. Additionally, high inhibition against the growth of *B. licheniformis*, *L. innocua*, *P. fluorescens*, *P. vulgaris*, and *P. putida* was seen [84]. Interestingly, various authors found that thymol was more effective in the inhibition of *E. coli* and *L. monocytogenes* growth than linalool [91,115]. This fact agrees with other findings reported, where the authors inform that thymol and carvacrol presented the highest antimicrobial activity, while linalool has moderate activity, and finally, p-cymene, borneol, and terpinenes presented weak inhibition activity [84]. In addition to the direct antimicrobial activity against several bacteria, yeast, and molds, this essential oil also has a strong antibiofilm activity, which prevents biofilm formation and allows for a reduction in synthetic antimicrobials [49].

On the other hand, clove essential oil presented high amounts of eugenol, which can be considered the main bioactive compound, and important contents of eugenyl acetate, β-caryophyllene, 2-methoxy-4-(2-propenyl)-phenol acetate, α-humulene, and α-caryophyllene [47,92]. This essential oil has significant inhibitory effects against the growth of *L. monocytogenes*, *S. aureus*, *E. coli*, *S Typhimurium*, *S. enterica*, and *C. jejuni*. The main antimicrobial effect of clove essential oil was related to eugenol, and the authors proposed that the lipophilic character of this molecule has the ability to partition the lipids of the membrane; to, thus, increase their permeability and decrease their integrity; to cause a disruption in proteins, to inhibition respiration, and to change ion transport [47]. 

Citral was reported as the major component in the lemongrass essential oil, with high amounts of geranial, neral, myrcene, limonene, cosmene, o-cimene, α-terpinolene, verbenol, citronellal, linalool, cis-carveol, nerol, atrimesol, carveol, geranyl acetate, and caryophyllene [93]. These compounds make this essential oil effective against *L. monocytogenes*, *Yersinia*, *E. coli*, *Staphylococcus*, *S. Typhimurium*, *L. plantarum*, *P. aeruginosa*, *B. cereus*, *B. subtilis*, *Enterococcus faecalis*, and *Enterobacter aerogenes* [93].

As was discussed, several compounds, mainly monoterpenes, terpenoids, and phenylpropanoids, are present in essential oils. A complex mixture of about 20–60 different compounds describes each essential oil, but normally, each of them is characterized by a high content of two or three compounds, and fewer amounts of the rest of the molecules. However, all of these molecules are vital for the essential oil bioactivity, since they may act in synergy and improve the antimicrobial properties of essential oil by acting at several sites of action at the cellular level [51,55]. Thus, the bioactivity of the essential oils, as occurs with plant extracts, cannot be attributed to a specific mechanism but can be explained by different interactions with microbials. In this sense, the hydrophobic interaction with the membrane cell constituents is vital and produces changes in membrane organization and cell membrane damage and, therefore, changes in fluidity, permeability, and release of intracellular material. Then, compounds forming essential oils also result in the depletion of intracellular ATP, and the changes in the electron transport chain and nutrient absorption cause important damages in the protein and DNA synthesis, dissipation of proton motive force, and inhibition of essential enzymes, which result in cell death [51,55,79]. 

Thyme essential oil in encapsulated and free form (0.05 and 0.1%) was tested against multiple microbial growths in beef burgers [116]. In this study, both free and encapsulated thyme essential oils produce a significant reduction in the counts of *Enterobacteriaceae* (between 1.15 and 1.79 log reduction in free form and between 2.19 and 3 log reduction in encapsulated form, after 8 days of refrigerated storage), total mesophilic counts (between 1.11 and 1.82 log reduction in free form and 1.58 and 1.99 log reduction in encapsulated form, after 8 days of refrigerated storage), *S. aureus* (between 1.07 and 2 log reduction in free form and between 4.13 and 4.69 log reduction in encapsulated form, after 8 days of refrigerated storage), lactic acid bacteria (between 0.72 and 1.26 log reduction in free form and between 1.44 and 2.04 log reduction in encapsulated form, after 8 days of refrigerated storage), and yeasts/molds (between 0.54 and 1.18 log reduction in free form and between 1.49 and 1.77 log reduction in encapsulated form, after 8 days of refrigerated storage). Additionally, the authors highlighted that the inhibitory effect was most evident with the use of encapsulated essential oil and attributed this effect due to the encapsulation process improving the life time of essential oils by protecting them against different degradations and evaporation. This is important not only to improve the shelf-life of the essential oil but also to improve their bioactivity and, thus, to maintain their antimicrobial activity until the end of meat products storage [116]. Moreover, thyme essential oil also presented a strong antifungal activity, which could be important in some meat products [117]. In a more recent study, the encapsulation of this essential oil using maltodextrin-casein as encapsulating agents showed a clear antimicrobial activity in burger-like meat products [118]. These authors conclude that this encapsulated essential oil effectively inhibits the growth of *S. aureus*, *E. coli*, *L. monocytogenes*, and *S. Typhimurium* tested in vitro and against thermotolerant coliforms (from 460 being the most probable number to 15 being the most probable number in control and treated samples, and after 14 days of storage) and *E. coli* (from 9.2 being the most probable number of complete inhibitions in the control and treated samples, and after 14 days of storage) in meat products. In another study, the authors evaluated the efficiency of thyme essential oils of two different varieties against four pathogenic microbial (*Escherichia coli*, *Salmonella typhimurium*, *Staphylococcus aureus*, and *Pseudomonas aeruginosa*) growth in minced beef meat [119]. In this case, the use of thyme essential oils produces a strong growth inhibition of both *E. coli* and *S. Typhimurium* at low concentrations (0.01–0.05%) (about 0.5 log reductions), while in high amounts (3%), the thyme essential oil produces a bactericidal activity against all microbial (from ~4–7 log reduction to non-detectable level). Additionally, the thyme essential oil extracted by supercritical fluids was used in the treatment of pork patties [90]. In this study, patties with essential oils were found to have significantly lower total plate (between 0.47 and 1.48 log reduction, depending on the treatment), lactic acid bacteria (between 0.33 and 0.56 log reduction), and *Enterobacteriaceae* counts (between 0.32 and 0.82 log reduction), which improve the microbial safety of this meat product [90].

Klein et al. [120] reported six essential oil combinations against bacteria, such as *Aeromonas hydrophila*, *Escherichia coli*, *Brochothrix thermosphacta*, and *Pseudomonas fragi–typical* bacteria in meat and meat products, at single-use and in combination. The mixture of essential oils, thymol, and carvacrol was the most effective combination. Similarly, various mixtures of six essential oils were recently proposed for the shelf-life enhancement of emulsion-based chicken sausages, and the authors found that the presence of clove and thyme in high amounts promoted the antimicrobial properties of the mixture [121]. 

Oregano and lavender essential oils, and their compounds carvacrol and linalool, were evaluated against *E. coli* and *S. aureus*. However, they just work against *S. aureus* [122]. In meat products, the addition of these herbs and spices and their essential oil extracts suppressed the growth of *Salmonella*, *Escherichia coli*, *Staphylococcus aureus*, *Listeria monocytogenes*, *Shigella flexneri*, and yeasts [42]. 

The *Listeria monocytogenes* inhibition in *Sous Vide* cook-chill beef was studied during storage at two refrigerates temperatures. The authors used both thyme and rosemary essential oils and concluded that rosemary essential oil is more effective at reducing the growth of this pathogen (1.23 and 2.63 log reduction at 2 °C and 8 °C, after 28 storage days, respectively) than thyme (0.57 and 0.06 log reduction at 0 and 28 storage days, respectively) at both storage temperatures [123].

The combination of oregano essential oil and radish powder was also proposed for the manufacture of nitrite-reduced fermented cooked sausages [28]. In this study, the addition of oregano essential oil to samples without nitrite controlled the growth of spoilage (mesophilic bacteria and coliforms) (similar to control samples with nitrite) and pathogenic (*Salmonella* spp., *Staphylococcus*, and sulfite-reducing clostridia) (not detected) microbials during 60 storage days. In view of these results, the authors concluded that this combination is a good strategy for reducing nitrite in meat products and, at the same time, ensuring their safety [28].

The addition of black pepper essential oil to fresh pork inhibits the growth of *Pseudomonas* spp. (2.1–3.05 log reductions) and *Enterobacteriaceae* (1.05–1.85 log reductions) during storage [124]. However, this essential oil did not produce any antimicrobial activity against lactic acid bacteria or *Brochothrix* spp. Additionally, the authors reported that, in this case, the application of black pepper essential oil produces a stronger inhibition against Gram-negative than Gram-positive bacteria, while its activity was dose-dependent. 

The essential oil obtained from Satureja was proposed for the manufacture of nitrite-reduced mortadella sausages [125]. In this study, the use of Satureja essential oil produces an inhibition of the *C. perfringens* growth during storage (30 days at 25 °C) (4.92 log reduction without nitrites, 5.32 log reduction in samples with essential oil +100 ppm nitrite, and complete inhibition in samples with essential oil +200 ppm nitrite). Moreover, the authors reported a synergistic effect between essential oil and nitrite, and thus, they proposed that Satureja essential oil could be used to produce meat products with reduced nitrite content since essential oil alone did not provide complete protection against this pathogen.

The reformulation of burger-like meat products with unencapsulated and encapsulated clove essential oil also resulted in interesting findings [92]. In this study, the addition of unencapsulated clove oil (at 3.047 mg/mL) to the burger formulation inhibited the *S. aureus* (0.86 log reductions) growth more efficiently than synthetic additive (nitrite), while encapsulated oil did not show this inhibitory effect [92].

Sage essential oil was used to extend the shelf-life of pork sausages [88]. This essential oil presented high antimicrobial activities against, *Salmonella* spp., *E. coli*, and *L. monocytogenes* (in all cases, pathogens were not detected). Moreover, mainly attributed to the antimicrobial activity of oxygenated monoterpenes, the use of sage essential oil also effectively reduced the aerobic mesophilic bacteria [88]. 

Coriander essential oil was added (0.075–0.150 µL/g) to nitrite-reduced cooked pork sausages, and the results obtained demonstrated a clear improvement in microbial quality [126]. In this study, the progressive reduction of nitrites and the addition of coriander essential oil demonstrated a clear inhibitory effect against microbial growth (total plate counts) (~0.2–0.5 log reductions), and the inhibitory efficiency was dose-dependent. 

Interestingly, the use of cinnamon essential oil in free form or encapsulated with chitosan as encapsulating agent reduces the microbial counts during beef patties storage [127]. These authors added different amounts (0.05 or 0.1%) of both types of essential oils to the patties formulation and concluded that, in both forms, free and encapsulated, the essential oil produced a strong inhibition of total mesophilic bacteria (1.19–2 log reduction with unencapsulated and 2.23–2.86 log reduction with encapsulated essential oil), lactic acid bacteria (~1–2 log reduction), *Enterobacteriaceae* (1.15–1.79 log reduction with unencapsulated and 3.23–3.86 log reduction with encapsulated essential oil), *Staphylococcus aureus* (2.12–2.62 log reduction with unencapsulated and 3.99–4.59 log reduction with encapsulated essential oil), and yeast/molds growth (1.62–1.92 log reduction with unencapsulated and 2.96–23.16 log reduction with encapsulated essential oil). Moreover, the results indicated that the antimicrobial effects of both are dose-dependent and more intense in the encapsulated essential oil [127]. 

Additionally, the use of cinnamon essential oil was proposed to control the microbial quality in emulsion-type sausages [128]. The authors concluded that the addition of this essential oil increases the shelf-life of vacuum-packed sausages (7 days in comparison with control), since it controlled the total counts and the lactic acid bacteria growth (0.54 log reductions) of the main bacteria associated with the spoilage of this type of products and reduced *Enterobacteriaceae* and fungi. 

Although their antimicrobial activity is interesting and promising, the cytotoxicity of some essential oils could limit their use in foods [49,79]. For example, the excessive consumption of lemongrass essential oil may cause tumors, gene damage, and carcinogenic effects [93], while the cytotoxic activity of thyme essential oil was related to the presence of *α*-pinene, borneol, and *β*-caryophyllene [129]. Moreover, essential oils obtained from *Salvia sclarea* L., *Cinnamomum camphora*, *Origanum compactum*, *Citrus aurantium Mentha piperita*, and *Eucalyptus globulus* were investigated for their toxicological constituents [112]. Thus, a complete determination and evaluation about their potential toxicity is a pre-requisite for their application on meat products as well as for the determination of the maximum safe dose. 

### 3.3. Edible Films and Coatings

Edible coatings are a thin layer of edible materials formed directly on the surface of the food and can be consumed with the food product [50,113,130]. They can be made with a variety of natural ingredients such as polysaccharides, proteins, and lipids. There is a growing market for natural antimicrobials used as preservatives in the form of edible films, as they can be used in different ways: sprinkled on meat or even dipping meat on them. They are harmless due to their natural potential [130,131].

Coating and edible films have been made of cellulose, starch (native and modified), pectin, seaweed extracts (alginates, carrageenan, and agar), gums (acacia, tragacanth, and guar), pullulan, and chitosan. These compounds can be used to extend the period of muscle foods by preventing dehydration, oxidative rancidity, and surface browning. When applied to wrapped meat products and exposed to smoke and steam, the polysaccharide film dissolves and becomes integrated into the meat surface, leading to higher yields, improved structure and texture, and reduced moisture loss.

A potential advantage of the incorporation of natural preservatives in the edible coatings and films is that it prevents the oxidative degradation of polyphenolic compounds, while a controlled and progressive release of bioactive compounds is achieved [50]. 

Milk protein-based edible films, which contain 1% pimento and 1% oregano, produce a high antimicrobial activity (against both, spoilage and pathogenic bacteria) during the storage of beef slices, and strong inhibition was observed in *Pseudomonas* spp. (0.95 log reduction) and *E. coli* O157:H7 (1.12 log reduction) [132].

Whey protein coatings containing antimicrobial agents such as oregano essential oil, 3-polylysine, or sodium lactate have been used in beef, in which, the development of lactic acid bacteria was completely reduced. The antimicrobial activity of carvacrol and cinnamon aldehydes, the main active compounds in the essential oils of cinnamon and oregano, were also evaluated. They were incorporated into edible films based on applesauce containing 1.5 and 3% carvacrol on chicken breast. These inactivated the decomposing microflora in chicken meat [131]. Similarly, the use of films containing 3% cinnamaldehyde and carvacrol produced a significant reduction in the growth *S. enterica* (4.3 log reduction), *L. monocytogenes* (2.2 log reduction), and *E. coli* O157:H7 (6.8 log reduction) of inoculated chicken breast [133], while in bologna-type sausages and ham, these films also produces a significant reduction in *L. monocytogenes* (1.4–3.3 log reduction in 3% carvacrol films and 0.35–1.20 log reduction in 3% cinnamaldehyde films) [134]. Moreover, the authors indicate that carvacrol presented higher antimicrobial effectiveness than films containing cinnamaldehyde, while the effect of films was higher in ham than in bologna sausages, indicating that meat constituents affected the antimicrobial efficacy.

On the other hand, edible chitosan-based films with thyme essential oil were tested to extend the shelf-life of cooked cured ham [135]. In this case, the use of edible films produced a decrease in aerobic mesophilic (2.57 and 2.61 log cycle reduction at day 7) and lactic acid bacteria (1.90 and 2.16 log cycle reduction at day 7) in slices of cooked cured ham during storage (21 days at 4 °C). Thus, the authors conclude that these edible films, based on chitosan and thyme essential oil effectively improved the shelf-life of this meat product [135]. In a similar way, a more recent study also proposed the use of thyme essential oil, in this case in combination with pomegranate peel extract to produce a chitosan–starch film [136]. The effectiveness of this film was tested on beef meat, and the use of this film (1% of pomegranate peel extract and 2% of thyme essential oil) produced the highest inhibition of *L. monocytogenes* growth (~2 log reduction after 21 days of storage at 4 °C) and reduced the total viable counts (~6 log reduction after 21 days of storage at 4 °C), lactic acid bacteria (5.21 log reduction after 21 days of storage at 4 °C), and *Pseudomonas* counts (6.13 log reduction) [136], which demonstrated that the application of this edible films in meat products could be an interesting strategy to ensure the safety of reformulated products. 

The development of antimicrobial films containing ground mustard seeds and packaging with low-fat ground beef has shown a potent increase in shelf life by 3.68 days, compared with regular beef, which just increases by 0.56 days. The differences between fat may improve the potential of food packaging [137]. Similarly, bologna-type sausages vacuum-packed with a polyvinyl polyethylene glycol film with mustard extracts produce a potent inhibitory effect against the growth of the pathogen *L. monocytogenes* (from 2.26 to 7.89 log reduction) and present an antimicrobial effect against lactic acid bacteria (4.46–7.38 log reduction), which extends the shelf-life of this meat product [138]. In packaged chicken meat, chitosan edible film with extracts of peppermint and Indian borage has bactericidal and antibiofilm activity against *E. coli* (>2.5 log reductions) and *Salmonella* spp. (~3 log reductions), and both active films inhibit the growth of both pathogens during refrigerated storage (15 days) [139]. 

The use of gelatin-starch edible film with corn stigma extract also extends the shelf-life of beef during storage, since it effectively inhibited the growth of mesophilic and psychotropic bacteria (1.2 log reduction) [140]. Similarly, the use of gelatin/agar-based films integrated with copper–zinc oxide nanoparticles and clove essential oil produces strong inhibition of the growth of *L. monocytogenes* and *E. coli* after 12 h of treatment [141]. Additionally, the same study observed a lower total aerobic bacterial count (~2.3 log reduction) in meat samples wrapped with this film, which demonstrated the potential use of this film as a potent antimicrobial.

There have been studies about lavender and oregano essential oils as gelatin films, against *E. coli* and *S. aureus*. The results indicate that both microorganisms exhibited sensitivity to all the active films [122]. Investigations of apple and tomato bio-pellicles containing 0.5–0.75% carvacrol or cinnamaldehyde protected the poultry against bacterial pathogens and deterioration and improved sensory properties. The efficacy of carvacrol-loaded polylactic acid film was assessed in ground beef with 12% and 5% fat. Carvacrol had a greater efficacy in meat with low-fat content, due to its solubility in fat limiting its interaction with the bacteria that grow in the watery portion of the meat [142].

In a very recent study, the authors developed chitosan edible films loaded with oregano and thyme essential oils [143]. These films produced a strong inhibition of the growth of psychrophilic bacteria (~4–5 log reduction), lactic acid bacteria (~3–4 log reduction), *Pseudomonas* (~4–5 log reduction), *E. coli* O157:H7 (~5–6 log reduction), *S. aureus* (~5–6 log reduction), and *S. Typhimurium* (~5–6 log reduction) in beef meat. This agrees with other authors, who affirm that the most promising strategy to inhibit the growth of pathogenic microbial is by coating meat and meat products with a film incorporated in essential oils [60].

Additionally, the use of chitosan-gelatin films with nanoemulsified garlic essential oil showed promising results in the microbial control of sliced omega-3 rich mortadella [52]. In this study, the authors observed that the application of the film retarded the growth of *L. monocytogenes* (~1–2 log reduction) and *P. aeruginosa* (~2–6 log reduction) in inoculated samples and against total coliforms (~2 log reduction), mesophilic (~2.5 log reduction), lactic acid bacteria (~3 log reduction), and psychrotrophic microbial (~3 log reduction). Thus, they conclude that this film can be used as a good preservative in reformulated meat products [52].

### 3.4. Bacteriocins

Bacteriocins are antimicrobial compounds coming from bacteria, which are peptides synthesized ribosomally and extracellularly released. They can be a destructive weapon against microorganisms, and the producing bacteria are not harmed since they have immunity proteins [144]. Bacteriocins are considered safe and non-active compounds, which are inactivated by digestive enzymes and, therefore, have little influence on the gut microbiota [144]. Gram-positive bacteria produce bacteriocins that can be classified into classes I, II, III, and IV. Classes I and II have low molecular weights (<10 kDa) and are thermostable, hydrophobic, and cationic peptides, but class II has an amphiphilic helical structure, promoting membrane depolarization and cell death [144]. Class III bacteriocins are large (>30 kDa) and heat-sensitive macromolecules (Sun et al., 2018), while class IV are not considered true bacteriocins due to their complex structural moieties (Kumariya et al., 2019). The antimicrobial action mechanisms of class I bacteriocins are related to the fact that these compounds cross the cell wall, inhibiting the lipids in the membrane and preventing the synthesis of vital components for the membrane, while the action mechanism of class II bacteriocins are based on the fact that they cross the cell wall and form pores in the cell membrane. Even so, there are bacteriocins (e.g., nisin) that exhibit both antimicrobial mechanisms [144]. The class II bacteriocins have strong antibacterial activity against *Listeria monocytogenes*, *Brochotrix* spp., *Clostridium* spp., *Bacillus* spp., and *Staphylococcus* spp. However, the main cause for limiting the use of some approved bacteriocins is the inability to inhibit Gram-negative pathogens [144]. Moreover, the use of bacteriocins in meat products did not produce any risk to human health due to the production of bacteria being GRAS, and they did not induce microbial resistance.

The most representative example of bacteriocins is nisin, from the lantibiotic group, produced by strains of *Lactococcus lactis* [25], effective against Gram-positive pathogenic strains, such as *Staphylococcus*, *Bacillus*, and *Clostridia*, and outgrowth spores from *Bacilli* and *Clostridia*. In addition, nisin is the only bacteriocin used commercially and recognized as a GRAS substance. This bacteriocin is considered a safe, natural, non-toxic and stable biopreservative [25]. However, there are other bacteriocins with potential use in the meat industry. In fact, lactic acid bacteria with bacteriocinogenic potential can produce a wide range of bacteriocins, including nisin, pediocin, pentocin, and sakacin [144]. Among all, *Lactobacillus sakei* and *L. curvatus*, typically found in meat products can produce sakacins and curvacins, respectively. Another commercial bacteriocins used as a food preservative is pediocin, produced by *Pediococcus acidilactici* [145,146].

The antimicrobial effectiveness of bacteriocins is highly dependent on the food constituents, such as the amount of salt or curing salts, fat, pH, enzymes, distribution into the food, or the interaction with other additives. Thus, the use of bacteriocins should be studied in real foods and against target bacteria. For example, the inefficiency of inhibiting important pathogens, the low solubility, and possible degradation due to enzyme activity are the main restrictions on the use of nisin in meat products [144]. 

The addition of nisin (0.001–0.013%) to vacuum-packed sliced cooked ham produces a significant reduction in mesophilic aerobic (3–4 log reduction) and lactic acid bacteria (2–3 log reduction) in comparison with control samples. Interestingly, the authors conclude that the lowest nisin addition showed similar shelf-life extension to the highest concentration; thus, minimal use of this bacteriocin can be satisfactory for meat industry purposes, with minimal technological and economic implications [147].

Similarly, in stored tray-packaged pork meat, the biopreservative effects of pentocin 31–1 (40 and 80 AU/mL) and nisin (75 AU/mL) were tested. These bacteriocins produced a clear inhibition of the growth of total bacteria (about 2 log reduction after 6 storage days), and the strongest inhibition activity was obtained by pentocin at higher amounts. Additionally, this bacteriocin also produces a significant reduction in the *Listeria* and *Pseudomonas* growth during chilled pork storage, which extends their shelf-life [148].

The bacteriocin BacFL31 (200 and 400 AU/g) was also tested as a preservative in ground turkey meat. This bacteriocin demonstrates an effective inhibition in the proliferation of *L. monocytogenes* (about 6 log reductions)*, S. aureus* (>2.5 log reductions), and *Salmonella typhimurium* (about 1 log reductions), as well as a strong inhibition of spoilage bacteria, including aerobic plate (about 3 and 6 log reductions for 200 and 400 AU/g, respectively), psychrotrophic (about 4 and 5 log reductions for 200 and 400 AU/g, respectively), and *Enterobacteriaceae* counts (about 2 log reductions) [149]. 

The antimicrobial effect of thyme essential oil, nisin, and their combination against *Escherichia coli* O157:H7 [150] and *Listeria monocytogenes* [151] was evaluated in minced beef (2.4% fat content). Nisin at a concentration of nisin at 500 or 1000 IU/g was effective against *L. monocytogenes* (1.15 and 3.57 log reductions, respectively) and had a synergistic effect when combined with thyme EO at 0.6% (complete inhibition to undetectable levels, ~9 log reductions). However, nisin was not able to control *E. coli* at the same concentrations. In addition, chitosan has been studied as an oxygen barrier in salami and in combination with lauric arginate and nisin, reducing the presence of *L. monocytogenes* in turkey, seafood, and fish [131].

Although the use of bacteriocins presented promising results, in a study, the addition of different amounts of nisin in combination with a mixed extract (green tea, stinging nettle, and olive leave extracts) in nitrite-free frankfurter sausages did not show any inhibition activity on total viable count or in yeast and mold growth [25].

The promising results of bacteriocins in meat products protection could be increased when a bacteriocin mix is used. This mix is more efficient since a bacteriocin can inhibit bacteria resistance to another specific bacteriocin [144]. Additionally, their stability despite changes in pH, the addition of salts, changes in temperature, and the addition of other additives makes them potential biopreservatives that ensure the safety of reformulated meat products, while reducing the use of synthetic or chemical additives.

## 4. Final Remarks

Demand for healthy, functional, and clean label meat products has increased in recent years as consumers are increasingly concerned about the harmful effects of fat, salt, or synthetic additive consumption. Several researchers have helped the meat industry in the search for alternatives to reduce the content of these controversial ingredients in their products. However, in general, the reduction of salts, fats, or nitrites can lead to greater microbial growth. Thus, the use of natural antimicrobials is a clean label alternative that is very effective in increasing the safety and shelf life of reformulated meat products. However, it is necessary to optimize the added dosage to obtain the maximum antimicrobial effect without affecting the sensory quality. Additionally, bioactive molecules can suffer chemical degradation during the meat products manufacture process, such as during high-temperature treatments (pasteurization, sterilization, baking), extrusion, or high-pressure treatments and can lose their activity [50]. 

In some cases, the addition of natural antimicrobials (e.g., essential oils) can produce undesirable changes in the sensory quality of meat products. For example, it was reported that rosemary essential oil was 0.2% [3], although this limit is dependent on the essential oil and the meat product. Moreover, the high volatility of essential oils also produces a fast loss of activity due to their loss to the environment [49]. Thus, to overcome this problem, several studies proposed their encapsulation, which controls the bioactive molecules released, increasing duration of antimicrobial activity, and reducing the intense odor of essential oils [152].

The application of natural preservatives into edible coatings is another promising strategy, since the release of bioactive compounds is progressive and controlled over time and produces a longer protection during food storage. 

Moreover, it is important to highlight that the antimicrobial effectiveness of these natural compounds is higher under “in vitro” conditions than in real foods [50], mainly due to the interaction of these bioactive molecules with fat, proteins, and minerals present in meat and meat products. Thus, is very important to prove the preservative effect of natural antimicrobials not only “in vitro” but also in real foods [51,53]. 

The legal considerations and regulations are another important aspect to consider to be able to apply these natural antimicrobials in the meat industry, and progress must be made in the study of their safety, while it is vital to create specific regulations for each one of them. Moreover, a common and uniform regulation among countries is necessary to regulate them, since legal aspects are completely different between administrations. Thus, currently, it is difficult to market reformulated meat products with natural antimicrobials throughout the world [79]. 

On the other hand, it is important to emphasize that it is of vital importance for natural antimicrobials to be used in the meat industry, that they be commercially available and at a “reasonable” price, and that their composition between different batches must be constant or very similar (for example, in essential oils or extracts, where there is great variation). In this sense, the processes for extracting and/or obtaining antimicrobial compounds must be standardized and completely controlled in order to achieve greater homogeneity in the final product, while the processes for obtaining them must be scalable at an industrial level in order to supply the commercial demand.

Furthermore, although sometimes antagonistic effects occur, it is important to note that the combination of two or more natural antimicrobials probably produces a more effective preservative effect in meat products and that the mechanisms of action against microorganisms are synergistic, ensuring the safety and quality of meat products. Thus, additional studies to assess the synergistic effect of different natural antimicrobials, to evaluate their optimum concentrations, and to assess their toxicity should be conducted. Even though potential green additives could be used in the reformulation of meat products, the promising results discussed in this manuscript are shown by the most natural antimicrobials. 

Finally, the findings reported by several studies, and condensed in this comprehensive review can be considered as a basis for a detailed investigation of the potential uses of natural antimicrobials in the meat industry.

## Figures and Tables

**Figure 1 foods-11-02613-f001:**
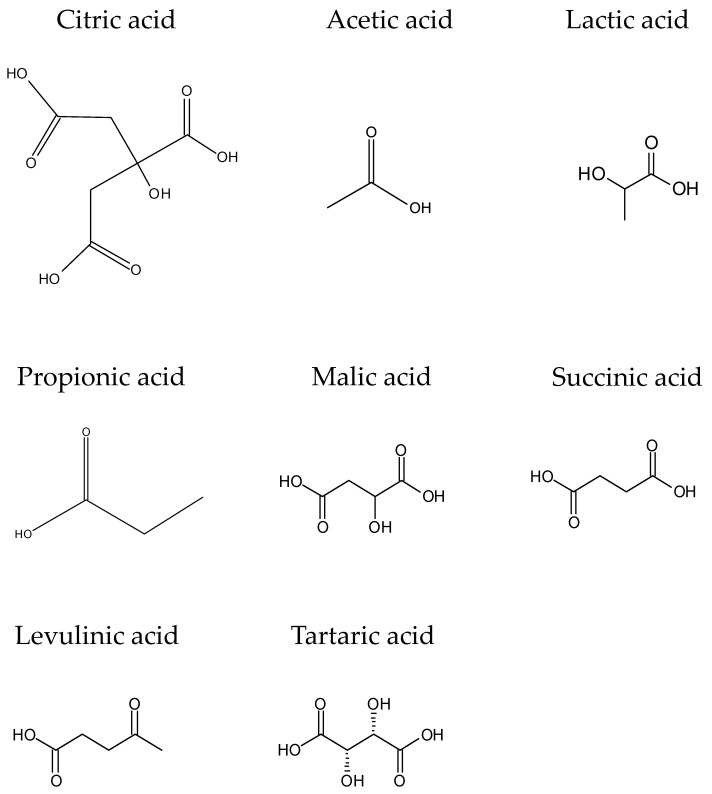
Chemical structure of organic acids commonly used in meat products.

**Figure 2 foods-11-02613-f002:**
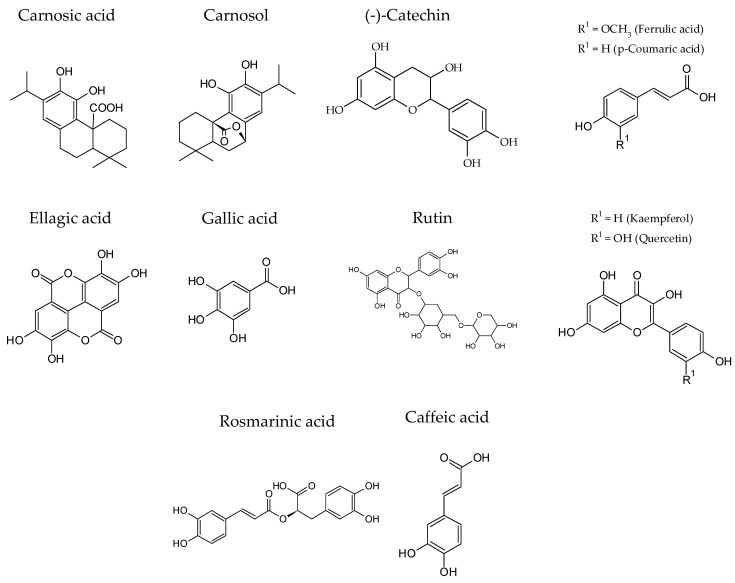
Chemical structures of the main bioactive compounds in plant-based extracts.

**Figure 3 foods-11-02613-f003:**
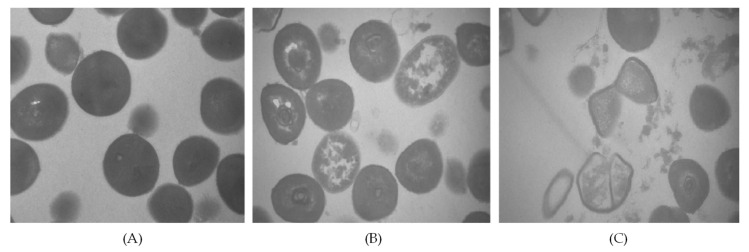
Example of *S. aureus* disintegration after the addition of 1 MIC (**B**) and 2 MIC (**C**) of *Amaranthus tricolor* extract in comparison with control samples (**A**) (obtained from Guo et al. [99]).

**Figure 4 foods-11-02613-f004:**
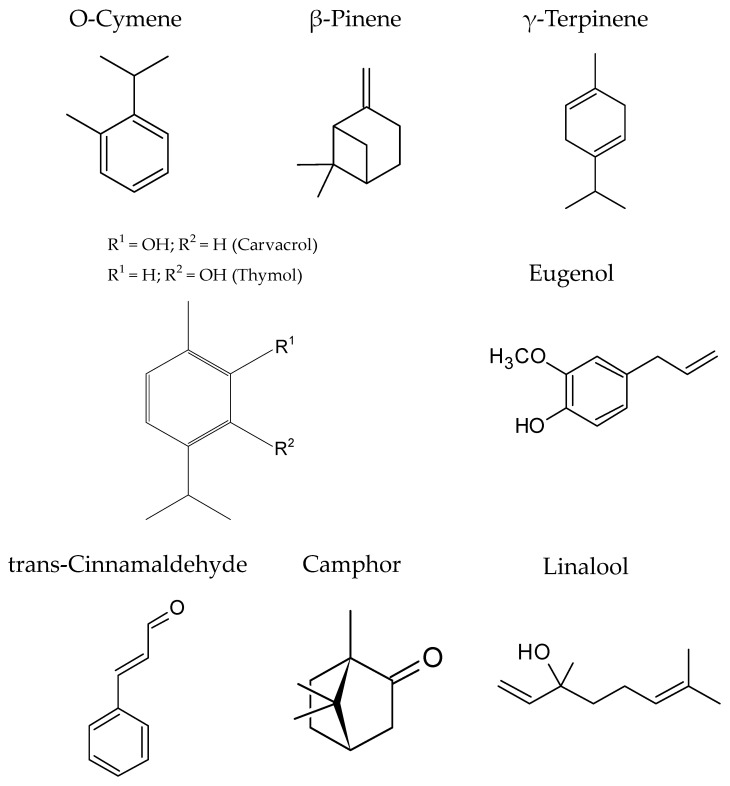
Chemical structures of the main bioactive compounds in essential oils.

**Table 1 foods-11-02613-t001:** Antimicrobial activity spectrum of organic acids and their salts.

Organic Acid or Salt	Antimicrobial Activity Spectrum
Lactic acid	*L. monocytogenes*, *S. aureus*, *E. faecalis*, *B. cereus*, *Salmonella* spp., *E. coli*, *P. aeruginosa*, *Proteus* spp., *C. albicans*, *S. cerevisiae*, *P. nordicum*, *P. purpurogenum*, *A. flavus*, *R. nigricans*, *Rhodotorula* spp.
Sodium lactate	Psychrotrophic bacteria, faecal streptococci, *L. monocytogenes*, *Enterobacteriaceae*, *E. coli, Salmonella* spp.
Potassium lactate	*L. monocytogenes*, *E. coli*, *Salmonella* spp.
Citric acid	*L. monocytogenes*, *S. typhimurium*, *E. coli* O157:H7, *A. flavus*, *P. purpurogenum*, *R. nigricans*, *F. oxysporum*, *S. cerevisiae*, *Z. bailii*.
Acetic acid	*L. monocytogenes*, *E. coli* O157:H7, *S. typhimurium*, *Enterobacteriaceae*, *P. nordicum*, *P. purpurogenum*, *A. flavus*, *R. nigricans*, *Fusarium* spp.
Propionic acid	*L. monocytogenes*, *E. coli*, *Salmonella* spp., *Cl. perfringens*, *A. flavus*, *Fusarium* spp., *Penicillium* spp., *R. nigricans.*
Tartaric acid	*S. typhimurium*, *A. flavus*, *Fusarium* spp., *Penicillium* spp., *R. nigricans*.
Formic acid	*E. coli*, *Salmonella* spp., *Cl. perfringens*, *A. flavus*, *Fusarium* spp., *Penicillium* spp., *R. nigricans*.
Sodium formate	*Streptococcus* spp., *Cl. perfringens*, *E. coli*, *S. enterica* *typhimurium*, *C. jejuni*.
Benzoic acid	*E. coli*, *L. monocytogenes*.
Succinic acid	*S. typhimurium*, *E. coli*, *B. subtilis*, *S. suis*.
Sorbic acid	*Fusarium* spp., *L. monocytogenes*, *E. coli*
Potassium sorbate	*Fusarium* spp., *L. monocytogenes*, *Salmonella* spp.

Adapted from Braïek and Smaoui [62].

## Data Availability

Not applicable.

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
