# Peer review of "Natural Antimicrobials: A Clean Label Strategy to Improve the Shelf Life and Safety of Reformulated Meat Products"

_foods, 2022, doi:10.3390/foods11172613_

Round 1

Reviewer 1 Report

General Comments:

The authors provide an extensive review of the use of many types and applications of natural antimicrobials in food. The document reads well, except for many English-related grammatical errors.  The main problem with the document is the repeated use of qualitative rather than quantitative indicators for inhibition of microorganism (see specific references below); these can easily be remedied by using log-reduction values to indicate the degree of inhibition. I don’t believe in the entire manuscript there is a single instance where the authors indicated that an antimicrobial produced x-log inhibition of a targeted organism; rather, they used vague terminology such as ‘inhibited, signification reduction, reduced at high levels’ which give no quantifiable degree of effect. The manuscript could be significantly improved if the authors were to include such data throughout.

Specific Comments:

1.       Line 52:  “…associated with arterial (?) diseases….”

2.       Line 60:  It is not clear what is meant by ‘technological advantages’;  please clarify.

3.       Line 62:  “Efforts for fat substitutes have been developed….”; ‘efforts’ does not fit in the context of the sentence.

4.       Line 68:  “Similarly, as with fat, the use of several strategies were….”

5.       Line 79: “….Clostridium botulinum and the germination of the heat-resistant spores….”

6.       Line 80: “…other multiple spoilage microorganisms….”

7.       Lines 81-95: The authors describes the use of vegetable extracts that contain nitrates which have been substituted for nitrite in fermented meat products as an example of alternate ingredients to reduce nitrite.  However, the vegetable-derived nitrates are converted to nitrite by fermentative organisms during meat fermentation.  In the USA, this had resulted in inconsistent levels of nitrite depending on the degree of fermentation and has been replaced by food ingredient companies performing fermentations with plant extracts, converting the vegetable nitrate to vegetable nitrite, and standardizing the subsequent extracts to defined levels of nitrite for use in meat products. This has resulted in removing the variability observed during in-product fermentation of vegetable nitrate to nitrite.

8.       Line 119:  “…pathogenic microorganisms are the main cause….”

9.       Line 119: “….losses for the meat industry, that also constitute…”

10.   Line 134:  “…..since these actions could increase contact between….”

11.   Line 137:  “Another important aspect is that many bacterial strains can bind to environmental or food contact surfaces and….”

12.   Line 149:  Not clear what the authors are trying to say:  “…..Gram-negative bacteria ( ) difficult the action(?) of some preservatives….”

13.   Line 158: “….the fat of meat products could surround the microorganisms and prevent the action….”

14.   Line 161:  Campylobacter and Salmonella are the most prevalent pathogens in chicken and turkey meat.”

15.   Line 175:  “….salt has an inhibitory effect against microorganisms by reducing….”

16.   Line 183: “…and changes the electron transport…”

17.   Line 184:  “….nitrites effectively inhibit the germination of spores of ….”.….but also other important microorganisms such as…..”

18.   Lines 221-234:  The authors should introduce the concept of isoelectric point (pI) of the organic acids in relation to pH of the food as it affects the equilibrium balance on the level of dissociation/association of the organic acid which affects it’s ability to penetrate bacterial cells (and then once the associated form of the organic acid is inside the bacteria, the cytoplasmic pH (~pH 7) is above the pI and the organic acid dissociates to produce the toxic anion form.

19.   Line 309:  “…strategy to extend the shelf life of meat products….”

20.   Line 313:  “…..with compounds found in the meat….”

21.   Line 331:  “…..to prolong the shelf life of meat and meat products…..”

22.   Line 336-338:  The sentence is not clear; please re-phrase.  “..the number of polyphenol-rich plants required by meat systems(?) ”

23.   Line 346:  “….plant extracts instead of the direct addition…”

24.   Line 454:  “….increase the shelf-life of meat products….”

25.   Line 463: “Frankfurter” does not need to be capitalized.

26.   Throughout: “Thymus essential oils” should be “thyme essential oils” (thymus is a gland).

27.   **Throughout the manuscript (too numerous to list), there is only qualitative assessment of antimicrobial activity in descriptive terms (significantly lower, shows activity, presents high antimicrobial activity, effective inhibition, highest antimicrobial activity, moderate antimicrobial activity, weak antimicrobial activity, stronger inhibition). There is no sense of degree of inhibition without providing log reduction numbers obtained by the various researchers that would indicate relative degree of effectiveness of the various antimicrobials.

Author Response

Reviewer #1 comments.

Comments and Suggestions for Authors

General comments:

The authors provide an extensive review of the use of many types and applications of natural antimicrobials in food. The document reads well, except for many English-related grammatical errors.  The main problem with the document is the repeated use of qualitative rather than quantitative indicators for inhibition of microorganisms (see specific references below); these can easily be remedied by using log-reduction values to indicate the degree of inhibition. I don’t believe in the entire manuscript there is a single instance where the authors indicated that an antimicrobial produced x-log inhibition of a targeted organism; rather, they used vague terminology such as ‘inhibited, signification reduction, reduced at high levels which give no quantifiable degree of effect. The manuscript could be significantly improved if the authors were to include such data throughout.

Response: The authors would like to thank the reviewer for their work and their comments to improve the manuscript. We have made the proposed changes and/or responded to the comments, and we hope that these changes may allow the article to be accepted for publication.

Specific Comments:

  1. Line 52: “…associated with arterial (?) diseases….”

Response: The reviewer is right. The sentence was corrected.

  1. Line 60: It is not clear what is meant by ‘technological advantages’;  please clarify.

Response: Following the reviewer´s comment, this statement was clarified.

  1. Line 62: “Efforts for fat substitutes have been developed….”; ‘efforts’ does not fit in the context of the sentence.

Response: The sentence was changed.

  1. Line 68: “Similarly, as with fat, the use of several strategies were….”

Response: The sentence was corrected according to the reviewer´s suggestion.

  1. Line 79: “….Clostridium botulinum and the germination of the heat-resistant spores….”

Response: The sentence was corrected according to the reviewer´s indication.

  1. Line 80: “…other multiple spoilage microorganisms….”

Response: The mistake was corrected.

  1. Lines 81-95: The authors describe the use of vegetable extracts that contain nitrates which have been substituted for nitrite in fermented meat products as an example of alternate ingredients to reduce nitrite. However, the vegetable-derived nitrates are converted to nitrite by fermentative organisms during meat fermentation. In the USA, this had resulted in inconsistent levels of nitrite depending on the degree of fermentation and has been replaced by food ingredient companies performing fermentations with plant extracts, converting the vegetable nitrate to vegetable nitrite, and standardizing the subsequent extracts to defined levels of nitrite for use in meat products. This has resulted in removing the variability observed during in-product fermentation of vegetable nitrate to nitrite.

Response: The reviewer is right. In fact, this is an important aspect. Thus, according to the reviewer´s suggestion, additional information was included in this paragraph about the pre-conversion of nitrates into nitrites using controlled fermentation.

  1. Line 119: “…pathogenic microorganisms are the main cause….”

Response: The sentence was corrected according to the reviewer´s indication.

  1. Line 119: “….losses for the meat industry, that also constitute…”

Response: The sentence was corrected according to the reviewer´s indication.

  1. Line 134: “…..since these actions could increase contact between….”

Response: The sentence was corrected according to the reviewer´s indication.

  1. Line 137: “Another important aspect is that many bacterial strains can bind to environmental or food contact surfaces and….”

Response: The sentence was corrected according to the reviewer´s indication.

  1. Line 149: Not clear what the authors are trying to say:  “…..Gram-negative bacteria ( ) difficult the action(?) of some preservatives….”

Response: Following the reviewer´s suggestion, this sentence was clarified.

  1. Line 158: “….the fat of meat products could surround the microorganisms and prevent the action….”

Response: The sentence was corrected according to the reviewer´s indication.

  1. Line 161: “Campylobacter and Salmonella are the most prevalent pathogens in chicken and turkey meat.”

Response: The sentence was corrected according to the reviewer´s indication.

  1. Line 175: “….salt has an inhibitory effect against microorganisms by reducing….”

Response: The sentence was corrected according to the reviewer´s indication.

  1. Line 183: “…and changes the electron transport…”

Response: The sentence was corrected according to the reviewer´s indication.

  1. Line 184: “….nitrites effectively inhibit the germination of spores of ….”.….but also other important microorganisms such as…..”

Response: The sentence was corrected according to the reviewer´s indication.

  1. Lines 221-234: The authors should introduce the concept of the isoelectric point (pI) of the organic acids in relation to the pH of the food as it affects the equilibrium balance on the level of dissociation/association of the organic acid which affects its ability to penetrate bacterial cells (and then once the associated form of the organic acid is inside the bacteria, the cytoplasmic pH (~pH 7) is above the pI and the organic acid dissociates to produce the toxic anion form.

Response: Following the reviewer´s suggestion, additional information about the isoelectric point was included in the text.

  1. Line 309: “…strategy to extend the shelf life of meat products….”

Response: The sentence was changed following the reviewer´s indication.

  1. Line 313: “…..with compounds found in the meat….”

Response: The sentence was changed following the reviewer´s indication.

  1. Line 331: “…..to prolong the shelf life of meat and meat products…..”

Response: The sentence was corrected following the reviewer´s indication.

  1. Line 336-338: The sentence is not clear; please re-phrase.  “..the number of polyphenol-rich plants required by meat systems(?) ”

Response: According to the reviewer's suggestions, the sentence was re-phrased.

  1. Line 346: “….plant extracts instead of the direct addition…”

Response: The sentence was corrected following the reviewer´s indication.

  1. Line 454: “….increase the shelf-life of meat products….”

Response: The sentence was corrected following the reviewer´s indication.

  1. Line 463: “Frankfurter” does not need to be capitalized.

Response: Thank you for your comment. The mistake was corrected.

  1. Throughout: “Thymus essential oils” should be “thyme essential oils” (thymus is a gland).

Response: The reviewer is right. The mistake was corrected throughout the text.

  1. **Throughout the manuscript (too numerous to list), there is only qualitative assessment of antimicrobial activity in descriptive terms (significantly lower, shows activity, presents high antimicrobial activity, effective inhibition, highest antimicrobial activity, moderate antimicrobial activity, weak antimicrobial activity, stronger inhibition). There is no sense of degree of inhibition without providing log reduction numbers obtained by the various researchers that would indicate relative degree of effectiveness of the various antimicrobials.

Response: The reviewer is right. Following the indications of the reviewer, the manuscript has been extensively modified and completed, including the degree of microbial reduction that each natural antimicrobial exerted. Thus, depending on the study, the log CFU/g of the control vs. treated samples and/or the logarithmic reductions and/or the reductions in the growth rate and/or the degree of inhibition (in percentage) have been included in the text and discussion.

Reviewer 2 Report

The aim of the manuscript is summarizing data on antimicrobials of natural origin and their effects on shelf-life extension and safety of reformulated meat products. If the collected literature data will critically be analysed and examined (especially parts dealing with chemistry), it can be accepted for publication (after extensive revisions).

Below there are specific comments/suggestions for the manuscript improvement.

Data summarized in Tables 1 and 2 do not bring any new information. Tables summarizing detailed data from studies on the use of natural agents in meat products preservation and shelf-life extension would be more innovative and interesting for readers.

Classification of compounds is misleading. Several plant genera such as Myroxylon and Styrax are producing resins and oleoresins containing substantial amounts of benzoic acid (Kokoska et al. 2019). This compound should therefore be considered as natural product (despite the fact that it is currently produced synthetically). The same applies for other organic acids.

Drawings of compounds should follow chemically correct and uniform style. Figures of certain compounds can be simplified (e.g. cinnamaldehyde) and many chemically related structures can merged (e.g. ferulic and p-coumaric acids, quercetin and kaempferol, thymol and carvacrol).

For comparison and critical evaluation of their efficacy, active concentrations of antimicrobial agents should be expressed in the same units.

Minor comments

Line 213. Cinnamon is not vegetable.

Line 369. …….against food borne patho-

Lines 767-770. One sentence should not be presented as single paragraph.

References

Kokoska L., Kloucek P., Leuner O., Novy P. 2019. Plant-derived products as antibacterial and antifungal agents in human health care. Current Medicinal Chemistry, 26(29): 5501-5541.

Author Response

Reviewer #2 comments.

Comments and Suggestions for Authors

General comments:

The aim of the manuscript is summarizing data on antimicrobials of natural origin and their effects on shelf-life extension and safety of reformulated meat products. If the collected literature data will critically be analysed and examined (especially parts dealing with chemistry), it can be accepted for publication (after extensive revisions).

Response: The authors would like to thank the reviewer for their work and their comments to improve the manuscript. We have made the proposed changes and/or responded to the comments, and we hope that these changes may allow the article to be accepted for publication.

Below there are specific comments/suggestions for the manuscript improvement.

Data summarized in Tables 1 and 2 do not bring any new information. Tables summarizing detailed data from studies on the use of natural agents in meat products preservation and shelf-life extension would be more innovative and interesting for readers.

Response: The authors understand your comment. In fact, we partially agree that it could be an option to make the tables with specific studies. However, this manuscript intends to offer general information, and not focus on specific studies. In addition, taking into account that the effectiveness of the application of an antimicrobial compound depends to a large extent on the compound itself, as well as on the meat product where it is applied (chemical composition, processing stages, etc.), the objective of these tables is to give an overview of the effect that each of the antimicrobials (and/or their main active compounds) can exert on pathogenic and spoilage microorganisms. This does not mean that in any of the studies, specifically, it may not have an effect, but most of the studies did present an inhibitory effect. Therefore, the authors consider it important to maintain the current tables in the text. However, if the reviewer deems it appropriate, we could remove table 1 and keep only table 2.

Classification of compounds is misleading. Several plant genera such as Myroxylon and Styrax are producing resins and oleoresins containing substantial amounts of benzoic acid (Kokoska et al. 2019). This compound should therefore be considered a natural product (despite the fact that it is currently produced synthetically). The same applies to other organic acids.

Response: Following the reviewer´s comment, important information about the genera Myroxylon and Styrax has been added. We have included relevant information and completed the text also with other lipophilic agents, including resins. However, although the specific mention of resins was added and discussed, it is important to note that they produce adverse reactions and sensitization (used topically) [e.g. https://doi.org/10.1111/cod.13263; https://doi.org/10.1111/ced.14266], which makes them invalid for use in the meat industry. In addition, resins, as a general rule, are not usually used as food ingredients due to their toxic potential. In fact, it was noticed that “Myroxylon-derived flavoring ingredients are not known to be used as a food” [https://doi.org/10.1016/j.fct.2019.110949], although in this case, they are GRAS. Anyway, following the reviewer's indications, we have included the study's information. We also include the reference proposed by the reviewer.

Drawings of compounds should follow chemically correct and uniform style. Figures of certain compounds can be simplified (e.g. cinnamaldehyde) and many chemically related structures can merged (e.g. ferulic and p-coumaric acids, quercetin and kaempferol, thymol and carvacrol).

Response: The reviewer is right. Figures were corrected according to the reviewer´s suggestion, and several compounds were simplified (cinnamaldehyde, acetic acid, propionic acid, levulinic acid, etc.), while others were merged.

For comparison and critical evaluation of their efficacy, active concentrations of antimicrobial agents should be expressed in the same units.

Response: According to the reviewer's indication, additional information about the concentrations of antimicrobial agents was specified in the text.

Minor comments

Line 213. Cinnamon is not vegetable.

Response: Following the reviewer´s indication, “cinnamon” was removed from the list.

Line 369. …….against food borne patho-

Response: The reviewer is right. The mistake was corrected.

Lines 767-770. One sentence should not be presented as single paragraph.

Response: Following the reviewer´s suggestion, the sentence was merged with the previous paragraph.

References

Kokoska L., Kloucek P., Leuner O., Novy P. 2019. Plant-derived products as antibacterial and antifungal agents in human health care. Current Medicinal Chemistry, 26(29): 5501-5541.

Response: The reference was included in the text, and resins and other lipophilic compounds were included in the discussion. In any case, it should be noted that it is not in line with the topic of the review, since it focuses on “human health care”, and not on its use as a preservative in the meat industry.

Reviewer 3 Report

The manuscript entitled “Natural antimicrobials: a clean label strategy to improve the shelf life and safety of reformulated meat products” has given a detailed review on natural anti-microbial extracts and ingredients which can be used to reduce the microbial load of meat products and can improve storage quality of meat products.

Overall the manuscript has covered a range of compounds and natural ingredients. However, the manuscript needs revision. Following points should be considered during revision:

1. The manuscript should be thoroughly checked for language and grammatical errors. Some of the authors are well renowned and very much capable to do so.

2. The paper has missed several natural preservatives and plant extracts which have been reported to reduce the microbial counts in meat products. Some of the papers which have reported different plant based ingredients with antimicrobial properties are mentioned below and should be discussed in the paper. This will improve the quality of the manuscript.

Author Response

Reviewer #3 comments.

Comments and Suggestions for Authors

General comments:

The manuscript entitled “Natural antimicrobials: a clean label strategy to improve the shelf life and safety of reformulated meat products” has given a detailed review on natural anti-microbial extracts and ingredients which can be used to reduce the microbial load of meat products and can improve storage quality of meat products.

Overall the manuscript has covered a range of compounds and natural ingredients. However, the manuscript needs revision. Following points should be considered during revision:

  1. The manuscript should be thoroughly checked for language and grammatical errors. Some of the authors are well renowned and very much capable to do so.

Response: The reviewer is right. According to the comments of reviewers #1 and #3, the manuscript was revised and language and grammatical errors were corrected.

  1. The paper has missed several natural preservatives and plant extracts which have been reported to reduce the microbial counts in meat products. Some of the papers which have reported different plant based ingredients with antimicrobial properties are mentioned below and should be discussed in the paper. This will improve the quality of the manuscript.

Response: We agree with the reviewer that there are countless studies, especially regarding natural preservatives derived from plants (extracts). We have tried to focus on the latest discoveries, but it is impossible to cover all the studies. We will be happy to include those studies that the reviewer considers important; however, although the reviewer indicates, "papers mentioned below", on the platform (susy, reviewer #3 report) do not appear article or any reference (see the image in the attached documment). Therefore, it is impossible for the authors to know which articles the reviewer is referring to. Please let us know the papers mentioned so we can make the changes you suggest.

Round 2

Reviewer 1 Report

The authors have made significant improvement to the manuscript by making use of prior suggestions of reviewers and I have no objections to publication, except the only things I noted were a couple of minor grammatical errors:

Line 160:  "...Gram-positive bacterial membrane...."

Line 371:  "....prolong the shelf-life of meat ....."

Other than that, again, I applaud the authors for the extra effort in improving their manuscript.

Reviewer 2 Report

Because the authors did not satisfactory address my comments, I do not recommend manuscript for publication.